# OpinioRAG: Towards Generating User-Centric Opinion Highlights from Large-scale Online Reviews

**Mir Tafseer Nayeem**    **Davood Rafiei**
Department of Computing Science
University of Alberta
{mnayeem, drafiei}@ualberta.ca

## Abstract

We study the problem of opinion highlights generation from large volumes of user reviews, often exceeding thousands per entity, where existing methods either fail to scale or produce generic, *one-size-fits-all* summaries that overlook personalized needs. To tackle this, we introduce OpinioRAG, a scalable, training-free framework that combines RAG-based evidence retrieval with LLMs to efficiently produce tailored summaries. Additionally, we propose novel reference-free verification metrics designed for sentiment-rich domains, where accurately capturing opinions and sentiment alignment is essential. These metrics offer a fine-grained, context-sensitive assessment of factual consistency. To facilitate evaluation, we contribute the first large-scale dataset of long-form user reviews, comprising entities with over a thousand reviews each, paired with unbiased expert summaries and manually annotated queries. Through extensive experiments, we identify key challenges, provide actionable insights into improving systems, pave the way for future research, and position OpinioRAG as a robust framework for generating accurate, relevant, and structured summaries at scale[1].

## 1   Introduction

Online reviews are an essential resource for consumers, offering firsthand insights into product quality, service reliability, and overall satisfaction across domains such as e-commerce, travel, and entertainment. With approximately 98% of online shoppers consulting reviews before making a purchase (PowerReviews, 2023), these user-generated evaluations shape consumer expectations, build confidence, and support informed decision-making (Foo et al., 2017; Gamzu et al., 2021). However, as online platforms expand, the sheer volume of reviews leads to *information overload* (Malhotra, 1984), making it challenging for users to extract meaningful insights. To cope with this overload, users typically skim a limited number of reviews—often fewer than ten—resulting in biased or suboptimal decisions (Kwon et al., 2015; Murphy, 2016). This highlights the pressing need for effective mechanisms that provide structured, personalized, and comprehensive access to relevant information.

A common approach to mitigating information overload is summarizing large collections of reviews into concise, digestible summaries (Hu & Liu, 2004; Pang & Lee, 2008; Suhara et al., 2020), distilling salient viewpoints while filtering out irrelevant content to help users quickly grasp essential insights (Ganesan et al., 2010; Hosking et al., 2024a). However, existing summarization methods have major limitations. First, prior work has primarily focused on summarizing short-form reviews, which pose relatively mild challenges given the capabilities of modern large language models (LLMs) (Zhang et al., 2024). Second, most approaches generate generic, *one-size-fits-all* summaries that fail to cater to personalized user needs. Consumers often seek context-specific insights aligned with their individual preferences—such as *room cleanliness*, proximity to *public transport*, availability of *fitness facilities*, or *pet-friendly* policies—but, current approaches lack the flexibility to generate query-specific summaries, limiting their utility for real-world decision-making. This work

---

[1] 🌐 ⬡ 🤗 **Project website:** `tafseer-nayeem.github.io/OpinioRAG`

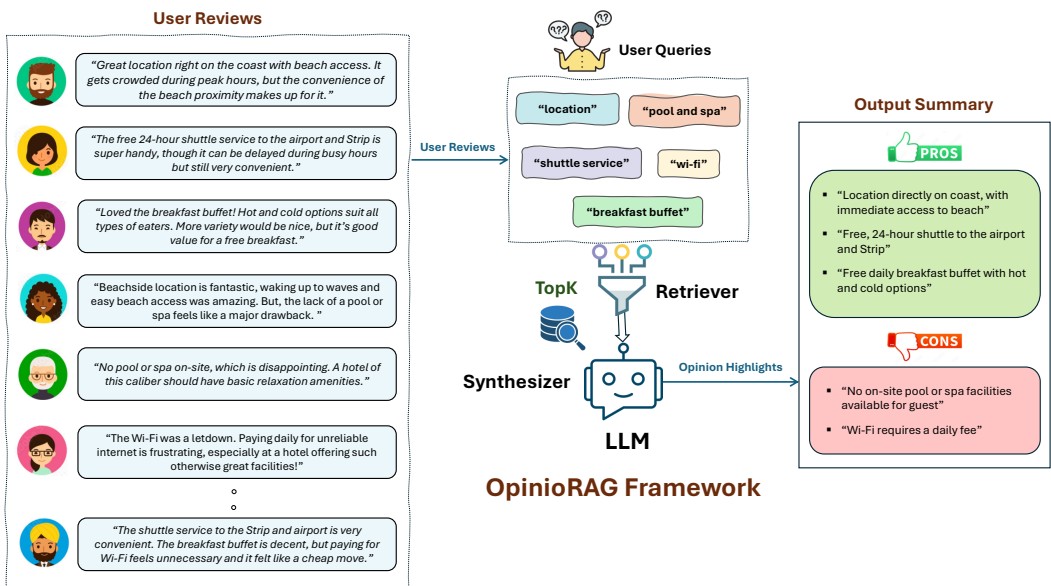

Figure 1: Overview of the `OpinioRAG` Framework. The framework comprises two stages: (1) **Retriever**—Extracts relevant sentences as evidence for each query from the user reviews, and (2) **Synthesizer**—Generates structured opinion highlights using LLMs conditioned on the retrieved sentences. The final output summary is a structured collection of opinion highlights addressing various user queries, providing an user-centric overview of reviews.

aims to bridge this gap by introducing a structured summarization framework that generates user-centric opinion highlights from large-scale, long-form reviews.

A significant challenge in developing such frameworks is the scarcity of annotated datasets that pair user reviews with summaries. To address this limitation, previous studies have employed self-supervised approaches to generate synthetic review-summary pairs by designating individual reviews as pseudo-summaries (Amplayo & Lapata, 2020; Elsahar et al., 2021). However, these datasets are limited in scale, typically containing at most 10 reviews per entity (Angelidis & Lapata, 2018; Chu & Liu, 2019; Bražinskas et al., 2020a), with only a few extending to hundreds (Angelidis et al., 2021; Bražinskas et al., 2021). Such scales are inadequate for real-world scenarios where entities often accumulate thousands of reviews.

To advance model development and evaluation in this area, we introduce a novel dataset—the first of its kind at this scale—designed to facilitate user-centric summarization from extensive long-form reviews. Our dataset comprises entities with over a thousand user reviews each, paired with unbiased expert reviews and manually annotated queries (Table 1). This resource serves as a benchmark for evaluating LLMs' ability to handle large-scale, noisy, and diverse inputs exceeding 100K tokens (Chang et al., 2024), providing a foundation for developing more sophisticated, user-centric opinion summarization models.

We further propose a training-free opinion highlight generation framework based on Retrieval-Augmented Generation (RAG) (Lewis et al., 2020), designed to effectively handle large-scale review corpora with user-centric queries (Figure 1). While long-context LLMs can process extensive input, they face limitations such as finite context windows, high computational costs (Ahia et al., 2023), privacy concerns from processing sensitive user information, and the *needle-in-a-haystack* problem (Liu et al., 2024; Karpinska et al., 2024), where critical information (e.g., *Wi-Fi quality*) is buried among irrelevant content. To estimate factual alignment, we introduce novel, reference-free verification metrics tailored for sentiment-rich domains like product and service reviews. Unlike existing metrics like `RAGAs` (Es et al., 2024) and `RAGChecker` (Ru et al., 2024), which focus on factual consistency in tasks such as question answering, our metrics capture nuanced opinions and sentiment polarity.

Our contributions—including a dataset, an opinion highlight generation framework, and verification metrics—offer a unique advantage over traditional RAG-based benchmarks,

which typically retrieve context documents from internet-scale repositories like Wikipedia and Common Crawl (Gao et al., 2024), sources often included in LLM pretraining corpora (Grattafiori et al., 2024; Groeneveld et al., 2024). This dependence on well-curated sources increases the risk of data contamination (Shi et al., 2024). In contrast, our method focuses on extracting insights from long-form, noisy user reviews, which are far less likely to overlap with LLM pretraining data due to rigorous filtering in pretraining pipelines (Penedo et al., 2024; Longpre et al., 2024). This setting provides a more realistic and challenging testbed for evaluating LLMs' robustness and generalization in real-world scenarios.

## 2 Related Work

**Opinion Summarization.** Opinion summarization has been widely studied through extractive and abstractive methods. Extractive approaches ensure factual consistency but often produce disjointed outputs (Angelidis et al., 2021; Li et al., 2023), while abstractive methods improve fluency but risk hallucination (Bražinskas et al., 2020b; Amplayo & Lapata, 2020). Recent work explores LLMs for summarization in zero- or few-shot settings (Bhaskar et al., 2023; Siledar et al., 2024a), but primarily on short-form inputs. In contrast, OpinioRAG targets long-form review corpora exceeding 100K tokens (Chang et al., 2024), and generates structured, query-specific summaries in a key-point format segmented into PROS and CONS.

**Benchmarks.** Existing datasets are limited in scale as they rely on pseudo-summaries constructed from short reviews (Amplayo & Lapata, 2020; Elsahar et al., 2021), and only a few scale to hundreds of inputs (Angelidis et al., 2021). Our OpinioBank is the first large-scale dataset pairing thousands of long-form reviews per entity with expert-written summaries and manually annotated queries, enabling evaluation under realistic user-facing scenarios.

**Query Focus and RAG.** Query-focused summarization enhances control and personalization (Xu & Lapata, 2020; Vig et al., 2022), with prior works focusing on general aspects from short reviews (Amplayo et al., 2021; Angelidis et al., 2021). OpinioBank instead offers fine-grained, entity-specific queries grounded in long-form content. Unlike prior RAG-based methods that lack scalability and factual verification (Hosking et al., 2024b), OpinioRAG employs a modular framework with structured verification metrics. While RAG has been applied in domains like finance, law, and dialogue (Zhao et al., 2023; Wang et al., 2025), OpinioRAG is the first to enable structured, query-specific summarization of long-form, noisy user reviews. For an extended discussion, see Appendix G.

## 3 OpinioBank: Construction and Analysis

We introduce OpinioBank—a large-scale, high-quality dataset designed to support user-centric opinion summarization from extensive long-form reviews. Unlike existing datasets focused on short-form reviews or synthetic review-summary pairs, OpinioBank comprises entities with over a thousand user reviews each, meticulously paired with unbiased expert reviews and manually annotated queries. This dataset serves as the first benchmark of its kind, aimed at advancing model development and evaluation for user-centric opinion summarization over large-scale, noisy, repetitive, and stylistically diverse inputs.

### 3.1 Data Sources

We selected hotel reviews as our primary domain due to their detailed, personalized narratives covering a wide range of user experiences. Moreover, hotels serve as an excellent case study due to the availability of expert summaries, although the same pipeline could be applied to other domains, such as product reviews or forum discussions.

**Source User Reviews.** Our user reviews were sourced from TripAdvisor[2], a widely-used platform integrating user-generated reviews with online travel booking services.

---

[2]https://www.tripadvisor.com

| Datasets | Domains | #Entities | #Queries | Unique #Queries | #Revs | #Sents | #Tokens | Book Len? | Expert Pairs | Meta data | P & C |
|---|---|---|---|---|---|---|---|---|---|---|---|
| MeanSum (2019) | Businesses | 200 | ✗ | ✗ | 8 | 41.1 | 561.01 | ✗ | ✗ | ✗ | ✗ |
| CopyCat (2020b) | Products | 60 | ✗ | ✗ | 8 | 30.38 | 463.62 | ✗ | ✗ | ✗ | ✗ |
| FewSum (2020a) | Businesses | 60 | ✗ | ✗ | 8 | 29.85 | 457.05 | ✗ | ✗ | ✗ | ✗ |
| OpoSum+ (2021) | Products | 60 | 240 | 4 | 10 | 71.8 | 1,194.0 | ✗ | ✗ | ✗ | ✗ |
| SPACE (2021) | Hotels | 50 | 350 | 7 | 100 | 910.58 | 16,770.18 | ✗ | ✗ | ✗ | ✗ |
| AmaSum (2021) | Products | 3,166 | ✗ | ✗ | 322.31 | 1,057.3 | 15,614.71 | ✗ | ✓ | ✗ | ✓ |
| ProSum (2024) | Restaurants | 500 | ✗ | ✗ | 6.70 | 71.34 | 1,236.38 | ✗ | ✓ | ✗ | ✗ |
| OpinioBank (*ours*) | Hotels | 500 | 5,975 | 1,456 | 1.5K | 10.5K | 207K | ✓ | ✓ | ✓ | ✓ |

Table 1: Comparison of our `OpinioBank` dataset with existing alternatives, focusing on long-form inputs (over 100K tokens) and user queries. #Entities denotes dataset size, #Queries refers to query count, #Revs indicates average reviews per entity, #Sents represents average sentences, and #Tokens indicates average tokens (using `GPT-4o` tokenizer) per entity. P & C stands for PROS & CONS. Other availabilities are indicated using ✓and ✗.

TripAdvisor reviews are notably longer than those found on other leading travel platforms, with an average length three times greater (D'Souza, 2024). This makes TripAdvisor an ideal resource for exploring the challenges of long-form summarization, especially with book-length inputs (exceeding 100K tokens) (Chang et al., 2024).

**Target Expert Reviews.** Annotated datasets pairing summaries with long-form reviews are scarce, as creating them manually is prohibitively expensive due to the extensive human annotation effort required to process lengthy and complex content (Tahaei et al., 2023). To address this challenge, we utilized Oyster[3], a platform specializing in professional hotel reviews based on firsthand, in-depth on-site evaluations conducted by expert reviewers (Oyster, 2015). These reviews are generated through a rigorous, multi-source assessment process involving online research, user review analysis, and expert evaluations. Each review on Oyster is structured into distinct sections, explicitly divided into 'PROS' and 'CONS', providing clear, unbiased assessments. We further refined these structured reviews, making them suitable as gold-standard summaries for our dataset.

## 3.2 Data Preparation

**Entity Pairing and Crawling.** To construct review-summary pairs, we identified 500 travel destinations from the Oyster platform. For each entity, we collected the overview section from Oyster, which contains critical summaries divided into 'PROS' and 'CONS'. To ensure accurate pairing, we searched for the same entities on TripAdvisor. In cases where multiple entities shared the same name, we employed unique identifiers such as hotel addresses and postal codes for disambiguation. Once matched, we crawled the relevant user reviews and corresponding expert reviews to build the dataset.

**Manual Query Annotation.** To support the design of our RAG framework, we conducted manual query annotation to assign the most relevant query term to each PROS and CONS sentence from the expert reviews. As a starting point, we used a predefined list of gold query terms from Pontiki et al. (2015), which corresponded to our target domain (i.e., Hotels). Leveraging this list as a reference, we manually annotated the sentences, adjusting query specificity as needed—either generalizing terms (e.g., "room," "location") or refining them (e.g., "room coffeemakers," "ocean views") to better align with review content. The annotation process ensured query diversity and minimized redundancy, thereby enhancing the dataset's utility and coverage (detailed dataset statistics in Table 6, a sample in Appendix A.1, and distribution of the entities presented in Figure 3).

**Review Alignment Verification.** Query terms served as a bridge between expert reviews and user-generated content. To ensure alignment, we validated the manually annotated queries against user reviews to confirm that summaries can be constructed. For each query, we searched for user review sentences containing exact matches of the query terms. If no

---

[3]https://www.oyster.com

exact match was found, the query was deemed non-applicable, and both the query and its corresponding sentence were removed from the expert reviews.

**Metadata Integration.** To enrich the dataset, we incorporate metadata from both the review text and the reviewer. From the *"review text"*, we include attributes such as `rating`, `helpful_votes`, and `publication_date`, while from the *"reviewer"*, we gather `user_reviews_posted`, `user_cities_visited`, and `user_helpful_votes`. These attributes provide valuable contextual signals related to user experience, credibility, temporal trends, and review quality, which can be utilized to enhance model scalability and performance (§7).

### 3.3 Comparative Analysis with Existing Datasets

**Dataset Coverage and Uniqueness.** Table 1 compares `OpinioBank` with existing opinion summarization datasets, highlighting its unique focus on long-form, book-length user reviews. Each entity in the dataset contains over a thousand reviews, offering a substantial volume of input texts. While `AmaSum` (Bražinskas et al., 2021) contains over three times the number of reviews as `SPACE` (Angelidis et al., 2021), its overall token count is lower due to domain differences—hotel reviews tend to be longer and more detailed than product reviews. We assess alignment with user reviews compared to widely used human-annotated opinion summarization datasets in Appendix A.2, and the results are presented in Table 7.

**Query Diversity and Specificity.** As shown in Table 1, our **OpinioBank** dataset exhibits significantly greater diversity in user queries compared to existing datasets such as `OpoSum+` (Amplayo et al., 2021) and `SPACE` (Angelidis et al., 2021). Prior datasets primarily focus on general aspects common to all entities, such as "location," "service," and "food," derived from short-form user reviews. In contrast, `OpinioBank` incorporates entity-specific queries that are uniquely tailored to individual entities, using long-form user reviews as input. Examples include "tuk tuk service," "museum access," and "yoga classes," which provide a richer and more nuanced representation of user needs and preferences (Table 6).

## 4 OpinioRAG Framework

Our proposed framework, **OpinioRAG**, builds on the `OpinioBank` dataset and combines the attributability and scalability of extractive RAG methods with the coherence and fluency of LLMs. It provides a scalable, training-free solution for generating user-centric opinion highlights from large volumes of user reviews, structuring them around specific user queries.

OpinioRAG decomposes the highlight generation task into *two sequential* stages: (1) **Retriever**—Extracts relevant sentences as evidence for each query, and (2) **Synthesizer**—Generates query-specific highlights in a desired style using LLMs conditioned on the retrieved sentences. As shown in Figure 1, the final summary is structured as a collection of opinion highlights for various user-centric queries related to a given entity, providing a comprehensive overview of a large collection of user reviews.

This dual-stage design offers advantages in terms of **(1)** `controllability` of the highlights based on user queries (e.g., *"fitness facilities"*), **(2)** `scalability` of the solution to large volumes of reviews while addressing LLMs' fixed context window constraints, **(3)** `modularity` of the approach, allowing flexible integration of different retrievers and diverse LLMs (varying in size, cost, or type), and **(4)** `verifiability` of generated highlights in a manageable units, allowing for fine-grained and context-sensitive assessments of factual alignment.

**Retrieval Stage.** We segment user reviews into individual sentences and use query-driven retrieval to extract the most relevant ones as evidence. This step reduces clutter by filtering key evidence before generation while ensuring comprehensive coverage of aspects from the source reviews. The retrieval process is formalized as:

$$S_Q = \text{Top-K}\left(\mathcal{R}(Q, D)\right) \tag{1}$$

where $Q$ represents the query set, $D$ is the corpus of user review sentences, $\mathcal{R}(Q, D)$ denotes the retrieval function, and $S_Q$ contains the Top-K retrieved sentences relevant to $Q$.

**Synthesizer Stage.**   The retrieved evidence is then utilized to generate query-specific highlights using LLMs, ensuring structured outputs in a predefined JSON format while adhering to desired key-point style (Appendix C). Formally, this process is represented as:

$$\text{Highlight}(Q) = \text{LLM}(Q, S_Q, C, \mathcal{E}, \mathcal{P}) \tag{2}$$

where $Q$ is the user query, $S_Q$ represents the set of Top-K retrieved sentences, $C$ defines stylistic guidelines, $\mathcal{E}$ consists of stylistic exemplars, and $\mathcal{P}$ denotes the prompt provided to the LLM (detailed in Figure 5 of Appendix).

### 4.1   Structured Verification in RAG via AOS Triplets

Our verification module builds upon prior work that established desiderata for human evaluation (Bhaskar et al., 2023) and aspect-level analysis at the summary level (Angelidis et al., 2021). Instead of relying on human judgments, we introduce three *novel*, automatic verification metrics that operate at the evidence-highlight level. The primary goal is to assess whether the LLM-generated highlights from the *synthesizer stage* are grounded in the retrieved user review snippets. Our explicit decoupling of retrieval and synthesis stages makes such targeted verification feasible (§4).

Our objective is to decompose sentences into structured components, enabling a fine-grained and systematic assessment of factual alignment. To achieve this, we employ Aspect-Opinion-Sentiment (AOS) triplets (Hu & Liu, 2004; Liu, 2012), utilizing an open-source model from Scaria et al. (2024). Our verification module applies the same triplet extraction process to both the retrieved evidence and the generated highlight, ensuring an interpretable and structured approach to verification (prompt in Figure 7 of Appendix). Since our metrics operate on an explicit alignment principle—where extracted aspects, opinions, and sentiments from the evidence and the generated highlight are directly compared—their validity is inherently derived from the structured nature of factual consistency rather than requiring correlation with human judgment. Additionally, our verification metrics are *reference-free* and *modular*, allowing the replacement of the triplet extraction model with more advanced alternatives as they become available. Each AOS triplet decomposes a sentence into *three* fundamental components:

- **Aspect** $a \in \mathcal{A}$: The attribute or feature being discussed (e.g., *"room bathroom"*).
- **Opinion** $o \in O$: The expression or judgment regarding the aspect (e.g., *"clean"*).
- **Sentiment** $s \in \{-1, 0, 1\}$: The polarity of the opinion (where $-1$ represents negative, $0$ neutral, and $1$ positive).

Let $R = \{r_1, r_2, \ldots, r_n\}$ denote the set of retrieved evidence sentences for a given query, and let $G$ represent the generated highlight. For each retrieved sentence and the generated highlight, we extract the corresponding AOS triplets:

$$\{(a_R^i, o_R^i, s_R^i)\}_{i=1}^n, \quad (a_G, o_G, s_G), \tag{3}$$

where $(a_R^i, o_R^i, s_R^i)$ represents the aspect, opinion, and sentiment extracted from the retrieved evidence $R$, and $(a_G, o_G, s_G)$ corresponds to the generated highlight. The factual consistency of the generated highlight is assessed based on three key criteria: Aspect Relevance (AR), Sentiment Factuality (SF), and Opinion Faithfulness (OF).

**Aspect Relevance (AR)**   This metric verifies whether the most frequently mentioned aspect in the retrieved evidence aligns with the generated highlight, ensuring aspect alignment helps maintain topical consistency and relevance. With the most frequent aspect among the retrieved sentences defined as:

$$a^* = \arg\max_{a \in \mathcal{A}} \text{freq}(a, R), \tag{4}$$

we define aspect relevance for an aspect $a$ in a generated highlight as

$$AR = 1\,(a^* = a_G), \tag{5}$$

where $1(\cdot)$ is an indicator function that returns 1 if the most frequent aspect appears in the generated highlight and 0 otherwise. The expectation $\mathbb{E}[AR]$, computed over all generated highlights, provides a measure of topical alignment.

---

**Evaluation Criteria**

| | |
|---|---|
| **Aspect Relevance (AR)** | Does the system summary cover the same topics or facets as the expert summary? |
| **Non-Redundancy (NR)** | Are aspects mentioned only once? Are key points repeated or paraphrased redundantly? |
| **Sentiment Agreement (SA)** | Is the tone (positive or negative) about aspects consistent between the summaries? |
| **Opinion Faithfulness (OF)** | Are the factual or evaluative claims in the system summary grounded in the expert summary? |
| **Overall Usefulness (OU)** | Would the system summary help a potential customer make a reasonable decision? |

Figure 2: LLM-as-a-Judge evaluation criteria used to assess the quality of the summaries.

**Sentiment Factuality (SF)**  This metric assesses whether the sentiment polarity in the generated highlight aligns with the predominant sentiment observed in the retrieved evidence. Neutral sentiments are excluded as they provide limited insight. For a given aspect $a$, the dominant non-neutral sentiment from the retrieved evidence is determined as:

$$s^* = \arg\max_{s \in \{-1,1\}} \ \text{freq}(s, R|a). \tag{6}$$

The sentiment factuality metric is then defined as:

$$SF = 1 \, (s_G = s^*) \,. \tag{7}$$

Similar to AR, the expectation $\mathbb{E}[SF]$, computed over all generated highlights, indicates if the sentiment polarity in the generated highlight is factually aligned with the predominant sentiment expressed in the retrieved evidence.

**Opinion Faithfulness (OF)**  This metric verifies how well the opinion in the generated highlight aligns with those extracted from the retrieved evidence. For a given aspect $a$ and sentiment $s$, let $o_G$ denote the opinion in the generated highlight, and let $\{o_R^i\}_{i=1}^{N_{a,s}}$ be the set of opinions extracted from the retrieved evidence that share the same aspect $a$ and sentiment $s$. A direct match between $o_G$ and any retrieved opinion is assigned a score of 1, while indirect matches are assessed using a semantic similarity function (e.g., cosine similarity). This allows semantically similar expressions (e.g., *"beautiful"* and *"stunning"*) to be considered faithful. The semantic alignment between $o_G$ and retrieved opinions, denoted as $OF$, is quantified as the expected similarity $\mathbb{E}[\text{Sim}(o_G, o_R^i)]$ across all retrieved opinions. The expectation then averaged over all generated highlights.

## 5  Evaluation

**Experimental Setup.**  To comprehensively evaluate the `OpinioRAG` framework, we conduct experiments involving multiple retrieval methods and LLMs. Both `OpinioRAG` and the long-context LLM baselines operate in a query-guided setting. In `OpinioRAG`, queries are used during the retrieval stage, whereas in long-context LLMs, they are directly included in the instruction prompt (Figure 6). In both settings, we compare the generated `PROS` and `CONS` for different queries against expert-provided summaries using automatic evaluation metrics (Appendix B.4). This evaluation measures content overlap with expert-written set of highlights (i.e., target summaries). We further analyze sentiment alignment with expert highlights in Section 6. Additional experimental details—including automatic metrics, retrievers, implementation setup, baselines, and other specifics—are provided in Appendix B, and sample output summaries are shown in Appendix I.

**Our Ablations.**  To examine the individual contributions of the **Retriever** and **Synthesizer** components within our `OpinioRAG` framework, we conduct ablation studies using BM25 and Dense retrieval methods. For each retriever, we select the top-ranked sentences for each query and merge them to form the final summary.

**LLM-as-a-Judge Evaluation.**  Recent advancements in LLM-based evaluation frameworks have demonstrated their potential as scalable and cost-effective alternatives to human evaluation (Li et al., 2024; Gu et al., 2025). To evaluate structured summaries organized into `PROS` and `CONS`—where each highlight is treated as an independent unit—we design

| Baselines | | PROS Scores | | | | CONS Scores | | | |
|---|---|---|---|---|---|---|---|---|---|
| | | R1 | R2 | RL | BS | R1 | R2 | RL | BS |
| Random | | 16.28 | 1.39 | 9.53 | 53.06 | 10.09 | 0.50 | 7.22 | 51.77 |
| Extractive Oracle | | 50.51 | 17.66 | 40.96 | 71.25 | 39.89 | 10.77 | 33.09 | 66.61 |
| TextRank | | 16.56 | 2.05 | 9.64 | 54.57 | 10.17 | 0.61 | 7.11 | 51.89 |
| LexRank | | 16.68 | 1.81 | 9.74 | 54.90 | 10.64 | 0.59 | 7.19 | 52.08 |
| **Model IDs** | **CL** | **Long-context LLMs** | | | | | | | |
| GPT-4o-mini | 128K | 29.97 | 5.76 | 17.21 | 64.95 | 18.97 | 2.70 | 12.22 | 59.94 |
| Claude-3.5-haiku | 128K | 32.70 | 7.03 | 19.30 | 67.37 | 20.07 | 3.03 | 13.44 | 61.41 |
| Gemini-2.0-flash | 1M | 30.62 | 5.75 | 17.87 | 65.45 | 20.81 | 3.73 | 13.70 | 60.92 |
| **Models/Ablations** | **Type** | **OpinioRAG (ours)** | | | | | | | |
| **BM25 (K=10)** | | 30.80 | 5.90 | 22.05 | 60.87 | 27.83 | 5.57 | 22.13 | 60.79 |
| └ GPT-4o-mini | 🔒 | 35.92 | 7.98 | 25.94 | 64.84 | 30.59 | 6.90 | 24.42 | 64.26 |
| └ Gemini-2.0-flash | 🔒 | 33.95 | 6.65 | 24.01 | 62.54 | 29.45 | 6.38 | 22.67 | 62.71 |
| └ Claude-3.5-haiku | 🔒 | 35.89 | 8.52 | 26.65 | 66.53 | 29.08 | 6.12 | 23.48 | 63.66 |
| └ Gemma-2-9B | 🔓 | 34.77 | 7.18 | 26.45 | 64.75 | **33.05** | **8.08** | **27.34** | 65.62 |
| └ Mistral-7B | 🔓 | 36.30 | 8.43 | 27.07 | 66.28 | 32.47 | 7.40 | 26.17 | 64.38 |
| └ Llama-3.1-8B | 🔓 | **37.51** | **9.13** | **27.41** | 66.62 | 32.61 | 8.06 | 25.79 | 64.79 |
| **Dense (K=10)** | | 28.86 | 4.99 | 20.77 | 61.91 | 25.37 | 4.64 | 20.11 | 60.82 |
| └ GPT-4o-mini | 🔒 | 35.69 | 7.66 | 25.96 | 65.55 | 29.48 | 6.70 | 23.52 | 64.19 |
| └ Gemini-2.0-flash | 🔒 | 33.97 | 6.58 | 24.16 | 63.42 | 28.74 | 5.90 | 22.23 | 62.94 |
| └ Claude-3.5-haiku | 🔒 | 35.27 | 8.05 | 26.19 | 66.76 | 27.52 | 5.18 | 22.37 | 63.58 |
| └ Gemma-2-9B | 🔓 | 34.45 | 6.56 | 25.86 | 65.14 | 32.31 | **8.32** | 26.84 | **65.81** |
| └ Mistral-7B | 🔓 | 36.33 | 8.20 | 26.97 | 67.19 | 31.38 | 7.09 | 24.94 | 64.24 |
| └ Llama-3.1-8B | 🔓 | 36.86 | 8.49 | 26.85 | 66.88 | 31.56 | 6.97 | 24.54 | 64.50 |

Table 2: Performance comparison of various models and retrieval methods (TopK = 10) in the OpinioRAG framework against baselines and long-context LLMs. The results are evaluated using lexical-based metrics (R1, R2, RL) and the embedding-based metric BERTScore (BS) for 'PROS' and 'CONS'. The icons 🔓 and 🔒 indicate open-source and closed-source models. **Bold** and underlined values denote the best and second-best results for each metric.

fine-grained, interpretable evaluation criteria and employ GPT-4o as an *LLM-as-a-Judge*. Each system-generated summary is assessed against expert summaries using a 5-point Likert scale across multiple quality dimensions, as illustrated in Figure 2. The detailed scoring rubric is provided in the Appendix (Figure 4).

## 5.1 Summarization Performance Evaluation

**Results & Analysis.** The results in Table 2 highlight several critical findings. Long-context LLMs struggle to retrieve and synthesize relevant information from extensive inputs, underscoring the challenges of handling large-scale reviews given user queries (Laban et al., 2024). Our ablation results reveal the importance of the synthesizer in OpinioRAG. While retrieval alone provides reasonable scores, the combination of retrieval and generation notably enhances performance across all metrics. BM25 consistently outperforms Dense retrieval, demonstrating its robustness in processing large-scale user reviews through effective lexical matching. Increasing the retrieval size from TopK = 5 to TopK = 10 consistently improves performance across models (TopK = 5 results are provided in Table 8 of Appendix), indicating the benefit of

| Model | Type | AR | NR | SA | OF | OU |
|---|---|---|---|---|---|---|
| **BM25 (K=10)** | | | | | | |
| Gemma-2-9B | 🔓 | 3.14 | 3.81 | 2.93 | 2.88 | 3.11 |
| Mistral-7B | 🔓 | 3.25 | 3.72 | 3.08 | 2.90 | 3.19 |
| Llama-3.1-8B | 🔓 | 3.26 | 3.85 | 2.98 | 2.86 | 3.19 |
| GPT-4o-mini | 🔒 | 3.17 | 3.57 | 2.90 | 2.86 | 3.12 |
| Gemini-2.0-flash | 🔒 | 3.29 | 3.46 | 2.93 | 2.91 | 3.18 |
| Claude-3.5-haiku | 🔒 | 3.24 | 3.70 | 3.04 | 2.92 | 3.19 |
| **Dense (K=10)** | | | | | | |
| Gemma-2-9B | 🔓 | 3.25 | 3.60 | 3.14 | 2.96 | 3.18 |
| Mistral-7B | 🔓 | 3.39 | 3.81 | **3.28** | 2.98 | **3.33** |
| Llama-3.1-8B | 🔓 | 3.32 | **3.89** | 3.13 | 2.95 | 3.25 |
| GPT-4o-mini | 🔒 | 3.31 | 3.69 | 3.10 | 2.96 | 3.25 |
| Gemini-2.0-flash | 🔒 | **3.42** | 3.45 | 3.15 | **3.02** | 3.32 |
| Claude-3.5-haiku | 🔒 | 3.38 | 3.81 | 3.14 | 2.99 | 3.31 |

Table 3: LLM-as-a-Judge evaluation results using BM25 and Dense retrievers with TopK = 10 configuration. **Bold** and underlined values denote the best and second-best results for each metric.

retrieving more evidence to capture dispersed information. Open-source models generally excel within our framework, highlighting the advantage of decomposing the task into retrieval and generation for better handling of extensive inputs.

| Models | TopK (K = 5) | | | | | | TopK (K = 10) | | | | | |
|---|---|---|---|---|---|---|---|---|---|---|---|---|
| | BM25 | | | Dense | | | BM25 | | | Dense | | |
| | AR | SF | OF | AR | SF | OF | AR | SF | OF | AR | SF | OF |
| GPT-4o-mini | 75.30 | 88.63 | 76.75 | 73.91 | 88.76 | 77.55 | 76.62 | 89.16 | 78.20 | 74.71 | **89.75** | 79.65 |
| Gemini-2.0-flash | **79.24** | 87.90 | 80.13 | 77.18 | 87.87 | 80.52 | 78.98 | 86.93 | 82.50 | 78.30 | 87.91 | **82.75** |
| Claude-3.5-haiku | 76.43 | 88.40 | 71.79 | 75.31 | 86.91 | 71.98 | 77.22 | 86.82 | 74.22 | 75.49 | 87.58 | 74.80 |
| Gemma-2-9B | 76.78 | 88.46 | 78.42 | 75.83 | 88.03 | 79.32 | 77.89 | 87.71 | 81.47 | 77.15 | 89.32 | 82.14 |
| Mistral-7B | 75.89 | 86.30 | 78.65 | 74.65 | 86.68 | 78.46 | 77.31 | 87.04 | 80.76 | 74.14 | 87.52 | 81.15 |
| Llama-3.1-8B | 77.65 | 87.45 | 78.34 | 73.99 | 87.21 | 79.80 | 78.82 | 87.40 | 81.11 | 74.28 | 88.91 | 82.35 |
| AVG. | 76.88 | 87.86 | 77.35 | 75.15 | 87.58 | 77.94 | 77.81 | 87.51 | 79.71 | 75.68 | 88.50 | 80.47 |

Table 4: Comparison of Aspect Relevance (AR), Sentiment Factuality (SF), and Opinion Faithfulness (OF) across various models using BM25 and Dense retrieval methods for TopK = 5 and TopK = 10. Results indicate that increasing TopK generally improves performance. BM25 is more effective for AR, while Dense retrieval performs better for SF and OF.

| Models | TopK (K = 5) | | | | TopK (K = 10) | | | |
|---|---|---|---|---|---|---|---|---|
| | BM25 | | Dense | | BM25 | | Dense | |
| | TPR | TNR | TPR | TNR | TPR | TNR | TPR | TNR |
| GPT-4o-mini | 83.75 | 52.98 | 87.13 | 54.35 | 84.86 | 54.46 | 87.61 | 55.34 |
| Gemini-2.0-flash | 80.04 | 54.57 | 82.52 | 56.49 | 79.44 | **58.02** | 81.97 | 57.14 |
| Claude-3.5-haiku | 82.88 | 53.86 | 84.81 | 55.45 | 82.38 | 55.12 | 83.97 | 56.27 |
| Gemma-2-9B | 86.50 | 52.87 | 88.28 | 54.24 | 86.45 | 54.30 | 88.36 | 55.99 |
| Mistral-7B | 89.71 | 45.32 | 92.12 | 47.18 | 90.79 | 44.94 | **92.48** | 46.41 |
| Llama-3.1-8B | 86.04 | 53.15 | 88.02 | 54.73 | 88.55 | 51.61 | 90.04 | 52.27 |

Table 5: Comparison of True Positive Rate (TPR) and True Negative Rate (TNR) across various models using BM25 and Dense retrieval methods with TopK = 5 and TopK = 10. Higher TPR indicates better alignment with positive aspects (PROS), while higher TNR reflects better alignment with negative aspects (CONS). The best results are highlighted in **bold**, while the second-best results are underlined and highlighted.

However, extracting critical drawbacks (CONS) remains challenging, with scores consistently lower than those for PROS. This observation aligns with prior findings that negative reviews are less frequent on online platforms (Venkatesakumar et al., 2021), and the *needle-in-a-haystack* problem is particularly evident when processing long-form inputs with long-context models (Karpinska et al., 2024; Kim et al., 2025). While our framework demonstrates promise, the performance gap between the Oracle baseline and our models indicates substantial room for improvement, particularly in identifying critical drawbacks.

Table 3 presents LLM-as-a-Judge evaluation results across five quality dimensions using TopK = 10 retrievals for BM25 and Dense retrievers. Open-source models such as Mistral-7B and Llama-3.1-8B exhibit strong performance, particularly in Non-Redundancy (NR), Sentiment Agreement (SA), and Overall Usefulness (OU). Proprietary models like Gemini-2.0-flash and Claude-3.5-haiku lead in Aspect Relevance (AR) and Opinion Faithfulness (OF), consistent with trends in Table 4. Dense retrievers generally outperform BM25 across all dimensions, with a similar trend noted in Table 9 for TopK = 5 retrievals.

## 5.2 RAG Verification Assessment

**Results & Analysis.** The results in Table 4 show that increasing retrieval size from TopK = 5 to TopK = 10 consistently enhances performance across all metrics, highlighting the benefit of retrieving more evidence. While BM25 generally excels in Aspect Relevance (AR), Dense retrieval demonstrates superior performance in Sentiment Factuality (SF) and Opinion Faithfulness (OF), effectively capturing nuanced sentiment and opinion relationships.

Nevertheless, SF remains challenging, likely due to LLMs' biases toward positive or polite language, which may downplay negative aspects or exaggerate complaints (§6). Additionally, the richer vocabulary of LLMs compared to noisy and unstructured user reviews may impact opinion alignment, as sophisticated language generation can diverge from simpler user-generated phrasing (case study of manual inspection in Appendix E).

## 6 Further Analysis: Sentiment Alignment of Highlights

We analyze the sentiment alignment between highlights generated by various LLMs from user reviews using our OpinioRAG framework, comparing them with reference expert highlights for different queries. Specifically, we evaluate the models' ability to accurately identify positive (PROS) and negative (CONS) highlights by measuring their alignment with expert highlights through **True Positive Rate (TPR)** and **True Negative Rate (TNR)**. TPR reflects the degree to which generated positive aspects align with expert highlights, while TNR measures the alignment of generated negative aspects. Sentiment classification is performed using SiEBERT (Hartmann et al., 2023), a model fine-tuned on diverse English datasets, including tweets and reviews, making it a robust choice.

The results in Table 5 reveal notable differences in model performance. Open-source models generally perform better at aligning positive highlights (PROS), while closed-source models excel at aligning negative highlights (CONS). This discrepancy may be attributed to the explicit safety tuning of open-source models aimed at reducing harmful or inappropriate content generation, which could inadvertently diminish their ability to accurately identify critical aspects (Jiang et al., 2023; Grattafiori et al., 2024). Dense retrieval consistently outperforms BM25 in capturing relevant PROS, indicating superior semantic matching, although BM25 shows slight advantages in TNR for certain models. Increasing the retrieval size from TopK = 5 to TopK = 10 improves both TPR and TNR, particularly in identifying PROS, suggesting that retrieving more evidence aids in capturing dispersed information. These findings underscore the importance of incorporating user-specific metadata to enhance the alignment between highlights and expert references. Reviewers with higher helpful votes or broader reviewing experience are likely to provide more credible insights (§7).

## 7 Discussion and Future Directions

**Leveraging Metadata for Review Selection.**   Our dataset includes rich metadata such as `review star ratings`, `helpfulness votes`, and `posting dates`, which can benefit both long-context LLMs and RAG-based methods. Helpfulness votes—accumulated over time, can provide a simple heuristic for identifying informative reviews; however, they may become outdated as temporary issues (e.g., broken facilities, hygiene concerns, or construction noise) are often resolved. Future work could incorporate *temporal reasoning* (Wang & Zhao, 2024) and develop functions that balance helpfulness scores with temporal relevance. Additionally, the lower 'CONS' scores observed across models (§5) indicate difficulties in accurately identifying significant drawbacks. Incorporating star ratings, as lower-rated reviews often emphasize negative aspects, could enhance the extraction of 'CONS' by prioritizing reviews most likely to highlight critical issues.

**Incorporating Opinion Holder Information.**   Online reviews vary in quality, ranging from informative insights to promotional content, spam, or manipulated reviews (Ott et al., 2013; Kennedy et al., 2019; Nayeem & Rafiei, 2023). Factors like a reviewer's experience, motives, and credibility significantly influence review quality (Feng et al., 2021; Gupta et al., 2020). Our dataset includes metadata such as `user_reviews_posted`, `user_cities_visited`, and `user_helpful_votes`, providing valuable signals on user expertise, credibility, and reviewing behavior. Incorporating these attributes can enhance the alignment between user reviews and expert summaries (we discuss the limitations in Appendix H).

## 8 Conclusion

We introduced OpinioRAG, a scalable, training-free framework that combines RAG-based retrieval with LLM synthesis to generate user-centric summaries from long-form reviews. Our proposed reference-free verification metrics offer robust assessment tailored to sentiment-rich domains. The OpinioBank dataset provides a comprehensive benchmark for assessing models on large-scale, noisy inputs. Extensive experiments demonstrate the effectiveness of OpinioRAG, offering valuable insights and establishing a strong baseline for future research.

## Acknowledgments

We thank the anonymous reviewers and the meta-reviewer for their valuable feedback and constructive suggestions to improve this work. This research was supported by the Natural Sciences and Engineering Research Council of Canada (NSERC). Mir Tafseer Nayeem also acknowledges support from the Huawei PhD Fellowship. Any opinions, findings, conclusions, or recommendations expressed in this material are those of the authors and do not necessarily reflect the views of the funding agencies.

## Ethics Statement

**Ethical Intent and Research Scope.** Our data collection process strictly adheres to ethical standards and is intended exclusively for non-commercial research purposes. All data obtained from TripAdvisor and Oyster is publicly accessible. Scraping such publicly available content does not violate the Computer Fraud and Abuse Act (CFAA), as no authentication or circumvention of access restrictions was involved.

**Licensing and Responsible Distribution.** All collected data complies with the Creative Commons Attribution-NonCommercial-ShareAlike 4.0 International License (CC BY-NC-SA 4.0)[4]. To promote transparency and responsible use, we release our dataset under the same licensing terms. Our OpinioBank dataset comprises only 500 expert-paired samples and is intended solely for evaluation and benchmarking. This limited scale minimizes both the volume of collected content and any potential commercial relevance.

**Technical Safeguards and Web Etiquette.** To prevent undue strain on source websites, we employed responsible scraping practices, including a controlled rate of data requests to avoid overloading servers. Our data retrieval process was designed to avoid interfering with site functionality or triggering denial-of-service (DDoS) protections. Furthermore, all HTTP requests included a standard User-Agent header, allowing server administrators to identify the nature of our automated access. We also consulted each website's robots.txt[5] file to ensure compliance with the Robots Exclusion Protocol (REP)[6].

**Privacy and Content Integrity.** To protect user privacy, we exclude all Personally Identifiable Information (PII), including reviewer IDs, usernames, real names, and location data. This ensures that the dataset remains anonymized and aligns with data protection regulations such as the GDPR and CCPA. As the dataset contains real-world user-generated content, some reviews may include personal details, subjective biases, or inappropriate language. We do not manually filter or edit review texts to preserve their authenticity and retain linguistic diversity for downstream research on user-generated content.

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

# Supplementary Material: Appendices

## A   OpinioBank: Dataset Description and Analysis

### A.1   Structure of an Entity in the Dataset

The dataset, `OpinioBank`, is organized in a standardized JSON format designed to ensure consistency and facilitate systematic evaluation of user-centric opinion summarization models. Each entity is represented by the following components:

---

**Entity Metadata**

**`entity_id` (Number):** A unique identifier assigned to each entity (e.g., hotels, restaurants).
**`entity_name` (String):** The name of the entity (e.g., "Fortune Hotel & Suites").

---

**Expert Review (Object)**

Structured expert reviews categorizing positive and negative aspects, organized according to annotated user-centric queries.

**Pros (Object)**

**Positive Attributes Associated with User Queries**
**Examples:**

- `"shuttle service"`: *"Free, 24-hour shuttle to the airport, Strip, and the Las Vegas Convention Center."*
- `"air conditioning"`: *"Effective air conditioning ensuring a comfortable indoor experience."*
- `"laundry and dry cleaning"`: *"Valet dry cleaning and laundry services available for guest convenience."*

**Cons (Object)**

**Limitations or Drawbacks Associated with User Queries**
**Examples:**

- `"room decor"`: *"Dated room decor that may not appeal to guests seeking modern aesthetics."*
- `"wi-fi"`: *"Wi-Fi requires a daily fee, which may be inconvenient for guests expecting complimentary access."*
- `"pool"`: *"No on-site pool available for guest relaxation."*

---

**User Reviews (Array of Objects)**

**Collection of User-Generated Reviews**
Each review is represented by the following attributes:

- review_id (Number): Unique identifier for each user review.
- text (String): The textual content of the user review, reflecting the user's experiences and opinions about the entity.
- rating (Number): User's numerical rating of the entity (e.g., on a scale from 1 to 5), indicating overall satisfaction level.
- helpful_votes (Number): Number of helpful votes received by the review, serving as a proxy for perceived usefulness to other users.
- publication_date (String): Date of publication (in the format YYYY-MM-DD), allowing for temporal analysis.
- user_reviews_posted (Number): Total number of reviews submitted by the user, providing insights into their reviewing behavior.
- user_cities_visited (Number): Number of cities visited by the user, reflecting their travel experience and potential expertise.
- user_helpful_votes (Number): Cumulative number of helpful votes received by all reviews written by the user, indicating their overall credibility and reliability as perceived by others.

---

The structured representation in OpinioBank effectively combines expert reviews and user-generated content, providing a robust foundation for evaluating models aimed at generating accurate, user-centric summaries from large-scale, long-form reviews. By organizing expert reviews around user-centric queries (e.g., *"shuttle service"*, *"air conditioning"*, *"laundry and dry cleaning"*), OpinioBank supports various research tasks, including query-based retrieval, sentiment analysis, temporal opinion analysis, and aspect-specific summarization. This comprehensive structure offers a versatile benchmark for developing sophisticated models capable of addressing real-world challenges in opinion summarization.

## A.2 Assessing Alignment with User Reviews

To assess alignment, we compared OpinioBank with widely used human-annotated opinion summarization datasets using an Extractive Oracle approach (Amplayo et al., 2021; Li et al., 2023)[7]. This method selects sentences with the highest ROUGE-L (RL) score for each reference sentence, providing an approximate upper bound. As shown in Table 7, OpinioBank demonstrates high overlap with user reviews, indicating that expert summaries are well-grounded in user experiences.

## A.3 Data Preprocessing

We did not apply any preprocessing to the user reviews or expert summaries in our dataset. User reviews are inherently noisy and exhibit diverse writing styles, varying lengths, redundancy, conflicting viewpoints, and inconsistent grammatical structures. In contrast, expert summaries from Oyster are polished, stylized, and vocabulary-rich, designed to provide clear and unbiased assessments. Our objective is to construct a robust and generalized system capable of effectively handling such complex, heterogeneous inputs and to evaluate LLMs' ability to generate accurate, user-centric opinion highlights that align with the desired styles. This approach ensures that our framework remains adaptable and applicable to real-world scenarios where unprocessed, diverse textual inputs are the norm.

---

[7]For consistency with the compared datasets, the 'PROS' and 'CONS' sections are merged into a paragraph-style format.

| Source User Reviews | | Target Summaries | | | |
|---|---|---|---|---|---|
| | | **Expert Reviews** | | **Annotated Queries** | |
| #Entities | 500 | #Entities | 500 | Total #Queries | 5975 |
| Avg. #Reviews | 1.5K | Avg. #Sents | 11.95 | Unique #Queries | 1456 |
| Avg. #Sents | 10.5K | Avg. #PROS | 8.30 | #Queries (Unigram) | 1721 |
| Avg. #Words | 196K | Avg. #CONS | 3.65 | #Queries (Bigram) | 3441 |
| Avg. #Tokens | 207K | Avg. #Tokens | 103.53 | #Queries (Trigram) | 640 |
| Max #Tokens | 975K | Max #Tokens | 190 | #Queries (>3 words) | 173 |

Table 6: Statistics of our `OpinioBank` dataset. The table presents detailed statistics of the dataset, covering both **Source User Reviews** and **Target Summaries**. For Source Reviews, we report the total number of entities ('#Entities'), the average number of user reviews per entity ('Avg. #Reviews'), average number of sentences ('Avg. #Sents'), words ('Avg. #Words'), and tokens ('Avg. #Tokens', calculated using the GPT-4o tokenizer) per entity. Additionally, we provide the maximum number of tokens ('Max #Tokens') found across all entities. For Target Summaries, we present statistics of **Expert Reviews** including average number of sentences ('Avg. #Sents'), positive sentences ('Avg. #PROS'), negative sentences ('Avg. #CONS'), and tokens ('Avg. #Tokens') per entity, as well as maximum tokens ('Max #Tokens'). The **Annotated Queries** section details the total number of queries ('Total #Queries'), unique queries ('Unique #Queries'), and the distribution of query lengths including unigram, bigram, trigram, and those containing more than three words.

| Datasets | Domains | Extractive Oracle | | |
|---|---|---|---|---|
| | | **R1** | **R2** | **RL** |
| 👤 **Human Annotations** | | | | |
| `FewSum` (2020a) | Businesses | 43.17 | 14.11 | 32.88 |
| `OpoSum+` (2021) | Products | 43.97 | 22.6 | 30.77 |
| `SPACE` (2021) | Hotels | 42.24 | 14.66 | 24.63 |
| ⌛ **Expert Review Pairings** | | | | |
| `OpinioBank` (*ours*) | Hotels | 52.41 | 19.74 | 45.02 |

Table 7: Extractive Oracle comparison across commonly used opinion summarization datasets. The table presents ROUGE-1 (R1), ROUGE-2 (R2), and ROUGE-L (RL) scores, where higher values indicate greater overlap between input reviews and target summaries. Our `OpinioBank` dataset, featuring expert review pairings, demonstrates superior alignment with user reviews.

# B  Experimental Details

Both the OpinioRAG framework and long-context LLM baselines are query-guided. In the OpinioRAG framework, queries are utilized during the Retrieval stage, whereas, for long-context LLMs, the queries are directly provided as input to the instruction prompt (Figure 6). For long-context baselines, we instruct the model to generate summaries divided into PROS and CONS. In contrast, the OpinioRAG framework generates query-specific highlights which are then categorized into PROS and CONS based on their alignment with expert sentiment (see Appendix A.1 for an example of the data structure).

In both settings, we compare the generated PROS and CONS for different queries with the expert-provided PROS and CONS using automatic evaluation metrics (Appendix B.4). This evaluation focuses on measuring content overlap with the set of expert highlights (a.k.a., target summaries). Additionally, we analyze sentiment alignment with the expert highlights in Section 6.

## B.1  Retrievers

We explore two retrieval strategies: BM25, a lexical retriever[8] that ranks document relevance based on term frequency-inverse document frequency (TF-IDF) (Robertson & Zaragoza,

---

[8]https://github.com/dorianbrown/rank_bm25

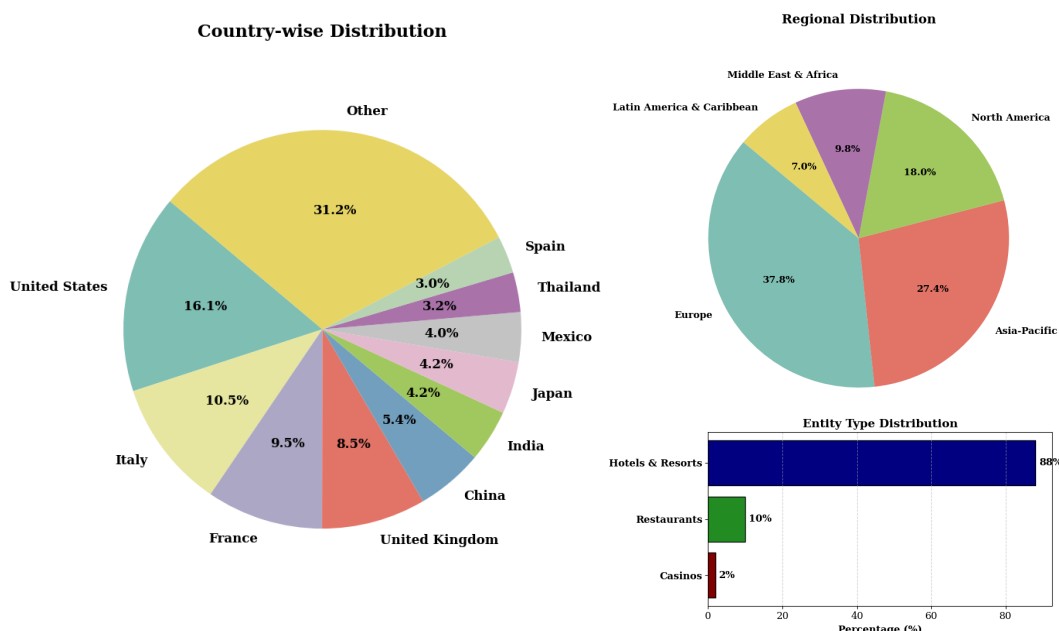

Figure 3: Geographical, Regional, and Entity Type Distributions in the `OpinioBank` Evaluation Dataset. The charts illustrate the distribution of entities across countries (left), regions (top right), and entity types (bottom right). Notably, the dataset comprises Hotels & Resorts (88%), Restaurants (10%), and Casinos (2%), providing a comprehensive overview of different categories for evaluation.

| Models/Ablations | Acc. | PROS Scores | | | | CONS Scores | | | |
|---|---|---|---|---|---|---|---|---|---|
| | | R1 | R2 | RL | BS | R1 | R2 | RL | BS |
| **BM25 (K=5)** | | 30.80 | 5.90 | 22.05 | 60.87 | 27.83 | 5.57 | 22.13 | 60.79 |
| └─ GPT-4o-mini | 🔒 | 35.74 | 7.73 | 25.56 | 64.78 | 30.29 | 6.91 | 24.14 | 64.10 |
| └─ Gemini-2.0-flash | 🔒 | 34.02 | 6.77 | 24.14 | 62.45 | 29.76 | 6.42 | 23.21 | 62.68 |
| └─ Claude-3.5-haiku | 🔒 | 35.19 | 8.19 | 26.10 | 66.20 | 29.02 | 6.43 | 23.60 | 63.98 |
| └─ Gemma-2-9B | 🔓 | 34.43 | 7.38 | 26.60 | 65.00 | 33.34 | **8.58** | **27.91** | **65.74** |
| └─ Mistral-7B | 🔓 | 35.97 | 8.30 | 26.86 | 66.46 | 32.89 | 7.89 | 26.69 | 64.55 |
| └─ Llama-3.1-8B | 🔓 | **37.11** | **8.77** | **27.38** | 66.44 | **33.56** | 8.27 | 26.73 | 65.31 |
| **Dense (K=5)** | | 28.86 | 4.99 | 20.77 | 61.91 | 25.37 | 4.64 | 20.11 | 60.82 |
| └─ GPT-4o-mini | 🔒 | 35.37 | 7.57 | 25.52 | 65.45 | 29.79 | 6.64 | 23.79 | 64.31 |
| └─ Gemini-2.0-flash | 🔒 | 33.99 | 6.48 | 24.22 | 63.55 | 29.14 | 6.01 | 22.45 | 63.06 |
| └─ Claude-3.5-haiku | 🔒 | 34.63 | 7.62 | 25.68 | 66.88 | 27.71 | 5.22 | 22.41 | 63.67 |
| └─ Gemma-2-9B | 🔓 | 33.74 | 6.51 | 25.80 | 65.03 | 32.19 | 7.99 | 26.91 | 65.71 |
| └─ Mistral-7B | 🔓 | 35.81 | 8.21 | 26.83 | **67.30** | 31.73 | 7.21 | 25.64 | 64.62 |
| └─ Llama-3.1-8B | 🔓 | 36.46 | 8.09 | 26.63 | 66.79 | 32.43 | 7.62 | 25.73 | 65.16 |

Table 8: Performance comparison of various models and retrieval methods (TopK = 5) in the `OpinioRAG` framework against baselines and long-context LLMs. The results are evaluated using lexical-based metrics (R1, R2, RL) and the embedding-based metric BERTScore (BS) for 'PROS' and 'CONS'. The icons 🔓 and 🔒 indicate open-source and closed-source models. **Bold** and underlined values denote the best and second-best results for each metric.

2009), and dense retrieval, which leverages neural embeddings to capture contextual semantics. This combination ensures both surface-level lexical matches and deeper conceptual relationships. For Dense retrieval, we employ Sentence Transformers (Reimers & Gurevych, 2019), specifically utilizing the `all-mpnet-base-v2` checkpoint, which demonstrates superior performance in semantic search across multiple benchmarks. For both BM25 and dense retrieval, the Top-K parameter specifies the maximum number of sentences (with a score

> 0.0) to be returned. For BM25, we apply a preprocessing pipeline to improve lexical matching, detailed in Appendix D.

## B.2 Long-context LLMs & Baselines

For long-context LLM baselines, the entire set of long-form user reviews for an entity are provided as input to the model. The LLMs are prompted with a task description, user queries, instructions, the expected number of 'PROS' and 'CONS' determined from the ground-truth reference, and a specified output format in JSON (prompt in Figure 6). If the context length exceeds the model's limit, older reviews are truncated, prioritizing more recent ones based on posting dates. Additional baselines are elaborated in Appendix F.

## B.3 Implementation Details

**Model Configuration:** For our `OpinioRAG` framework, we utilize both open-source models (`Mistral-7B`, `Llama-3.1-8B`, `Gemma-2-9B`) and closed-source models (`Claude-3.5-haiku`, `GPT-4o-mini`, `Gemini-2.0-flash`). We apply consistent hyperparameters across all models: `max_new_tokens` = 256, `temperature` = 0.7, and `top_p` = 0.9.

For the long-context baselines, we retain the default parameters of the long-context LLMs (`Claude-3.5-haiku`, `GPT-4o-mini`, `Gemini-2.0-flash`), except for setting `max_tokens` = 512, which ensures the models can generate summaries effectively for long-form user reviews.

**Semantic Alignment in Opinion Faithfulness:** To measure the degree of semantic alignment, we use a pre-trained model fine-tuned on restaurant reviews[9] to obtain contextual embeddings and compute cosine similarity between the generated and retrieved opinion vectors. This approach allows semantically similar expressions (e.g., "beautiful" and "stunning") to be treated as faithful matches. The cosine similarity score quantifies the alignment between opinions, enabling a graded assessment of semantic consistency.

## B.4 Evaluation Metrics

We evaluate model performance using both lexical and embedding-based metrics. Specifically, we compute F1 scores for ROUGE-1, ROUGE-2, and ROUGE-L (R1, R2, RL) (Lin, 2004) and BERTScore (Zhang* et al., 2020), following established practices in opinion summarization (Bhaskar et al., 2023). Although ROUGE has recognized limitations (Tay et al., 2019; Shen & Wan, 2023), we report these scores to ensure comparability with previous studies (Bhaskar et al., 2023; Lei et al., 2024; Siledar et al., 2024b; Hosking et al., 2024b), benchmark our dataset, and evaluate various models within our framework. Due to the structured and stylistic nature of our keypoint highlights, conventional coherence and fluency metrics (Bražinskas et al., 2020a; Angelidis et al., 2021) are less suitable for the output summaries (Appendix C).

| Model | Type | AR | NR | SA | OF | OU |
|---|---|---|---|---|---|---|
| **BM25 (K=5)** | | | | | | |
| Gemma-2-9B | 🔓 | 3.06 | 3.85 | 2.90 | 2.83 | 3.03 |
| Mistral-7B | 🔓 | 3.18 | 3.87 | 2.99 | 2.87 | 3.13 |
| Llama-3.1-8B | 🔓 | 3.16 | 3.89 | 2.90 | 2.81 | 3.11 |
| GPT-4o-mini | 🔒 | 3.11 | 3.68 | 2.85 | 2.81 | 3.06 |
| Gemini-2.0-flash | 🔒 | 3.22 | 3.53 | 2.88 | 2.89 | 3.12 |
| Claude-3.5-haiku | 🔒 | 3.20 | 3.83 | 2.95 | 2.88 | 3.16 |
| **Dense (K=5)** | | | | | | |
| Gemma-2-9B | 🔓 | 3.18 | 3.74 | 3.01 | 2.88 | 3.12 |
| Mistral-7B | 🔓 | **3.36** | 3.94 | **3.19** | 2.94 | **3.30** |
| Llama-3.1-8B | 🔓 | 3.25 | **3.95** | 3.06 | 2.88 | 3.19 |
| GPT-4o-mini | 🔒 | 3.20 | 3.80 | 3.00 | 2.90 | 3.16 |
| Gemini-2.0-flash | 🔒 | 3.32 | 3.54 | 3.06 | **2.98** | 3.22 |
| Claude-3.5-haiku | 🔒 | 3.30 | 3.88 | 3.08 | 2.94 | 3.23 |

Table 9: LLM-as-a-Judge evaluation results using BM25 and Dense retrievers with TopK = 5 configuration. **Bold** and underlined values denote the best and second-best results for each metric.

## B.5 JSON Parsing

To ensure that the LLMs output in a structured JSON format, we employ several strategies. These include explicitly stating the requirement for JSON output in the prompts, providing

---

[9]https://huggingface.co/saitejautpala/bert-base-yelp-reviews

a sample JSON structure, and incorporating in-context examples with the desired format. For models such as those from OpenAI[10] (`GPT-4o-mini`), we specify formatting instructions by configuring the necessary fields and descriptions (e.g., `response_format=''type'':` `''json_object''`). Similarly, for `Gemini-2.0-flash`, we use field descriptions (e.g., `generation_config=''response_mime_type'': ''application/json''`) to enforce JSON outputs, ensuring reliable evaluation.

To effectively parse structured outputs from LLM outputs, we employ a robust and flexible JSON extraction pipeline designed to handle various formatting inconsistencies. The parsing process is divided into two primary strategies:

**Direct JSON Parsing.**  We first attempt to parse the entire output text directly as JSON using standard JSON libraries. This approach ensures the most efficient handling of well-structured responses without additional processing.

**Fallback JSON Extraction.**  When direct parsing fails, we proceed with a multi-step fallback approach. This includes:

- **JSON Block Extraction:** Using regular expressions to identify and isolate JSON-like structures from the surrounding text, especially when the LLM's output contains additional commentary or formatting artifacts.
- **Pattern Matching:** Applying specific patterns to extract relevant keys (e.g., `pros` and `cons` or `review`) from the detected JSON block.
- **Validation:** Attempting to parse the extracted JSON block to confirm its validity. If valid, the content is returned; otherwise, further processing is applied to refine the extraction.

### B.6 System Message Design

To guide the LLMs, we developed a system message specifying the model's role and constraints. The message defines the LLM as an "expert summarizer of user reviews" within the domain of "hotels and restaurants," with a specialization in "travel." These elements were designed with several key considerations:

**Role and Task:** Defining the LLM as an expert ensures focused, high-quality outputs. It helps the model capture relevant sentiments and aspects while minimizing irrelevant details.

**Domain:** Narrowing the scope to hotels and restaurants ensures the model prioritizes key factors such as service quality, location, and amenities—critical in user-generated travel reviews.

**Specialization:** Adding a travel specialization refines the model's focus on aspects unique to travelers, such as proximity to attractions and comfort during stays.

> **System Message**
>
> You are an **expert summarizer** of user reviews for **hotels and restaurants**, specializing in **travel**!

## C   Key-Point Style Text

Key-point style text organizes information into concise, easily interpretable bullet points or structured lists, enhancing readability by allowing users to swiftly identify relevant details. This approach is particularly effective for summarizing user reviews, where clarity, relevance, and brevity are essential. Our `OpinioRAG` framework leverages this style to produce structured, user-centric opinion highlights that are both accessible and interpretable (Bar-Haim et al., 2020; 2021).

---

[10]Structured Outputs API, released on August 6th, 2024.

**Characteristics of Key-Point Style Text**

- **Conciseness**: Presents information succinctly, avoiding redundancy and providing focused insights.

- **Clarity**: Clearly separates distinct aspects, enhancing interpretability and coherence.

- **Scan-Friendly**: Facilitates efficient information retrieval, enabling users to quickly pinpoint relevant details.

- **Structured**: Organizes information hierarchically, improving readability and ensuring logical flow.

**Examples of Key-Point Style Text.** The following examples demonstrate how the key-point style text format is applied within the context of hotel reviews, organized by user-centric queries.

---

**Example: Positive Key-Points (PROS)**

- **"room amenities"**: Spacious rooms with premium linens, modern bathrooms, and high-quality toiletries.
- **"hotel location"**: Convenient access to major attractions and public transportation.
- **"staff service quality"**: Attentive and friendly staff providing prompt assistance.
- **"food quality"**: Delicious breakfast options and diverse menu selection.

---

**Example: Negative Key-Points (CONS)**

- **"room condition"**: Furnishings appeared outdated and not well-maintained.
- **"internet access"**: Internet access requires an additional fee.
- **"fitness center availability"**: Lack of fitness center and pool availability.
- **"noise level"**: Noticeable street noise even at higher floors.

---

The adoption of key-point style text directly supports the goals of the `OpinioRAG` framework. By organizing generated summaries into distinct points, the framework ensures high relevance and interpretability, particularly when addressing diverse user queries. Additionally, the clarity and modularity of key-point style text enable efficient comparison between generated summaries and expert references, thereby improving evaluation accuracy and reliability.

## D   Preprocessing for BM25 Retrieval

To enhance the retrieval quality of our `OpinioRAG` framework when using **BM25**, we apply a specialized preprocessing pipeline aimed at improving lexical matching between user queries and user review sentences, similar to previous pipelines (Nayeem & Rafiei, 2024; Guanilo et al., 2025). As BM25 is a lexical retrieval method, effective text normalization is essential for better matching accuracy. Importantly, this preprocessing is exclusively applied to **BM25 retrieval** and is only used for calculating relevance scores during the retrieval stage. The original, unprocessed sentences are preserved for the **Synthesizer stage** before being inputted to the LLMs. In contrast, **Dense retrieval** directly processes the original sentences without any preprocessing.

**Text Normalization.**  We begin by converting all text to lowercase to maintain consistency. HTML tags, URLs, and email addresses are removed using regular expressions to eliminate irrelevant content. Non-ASCII characters are also discarded to focus on standard textual information.

**Contraction Expansion.**  To improve compatibility with lexical matching, contractions are expanded using the `contractions` library[11]. For example, `"can't"` is converted to `"can not"`, ensuring a more comprehensive word matching process.

**Token Protection.**  Certain special tokens such as '`24-hour`', '`24/7`', and '`wi-fi`' are temporarily replaced by placeholders to prevent them from being incorrectly processed. For instance, '`wi-fi`' is converted to '`SPLWIFI`' during preprocessing and reverted to '`wifi`' after processing.

**Punctuation and Digit Removal.**  All punctuation marks and digits are replaced with spaces, retaining only alphabetic characters and whitespace. Additionally, hyphens and slashes are replaced with spaces to enhance tokenization consistency.

**Lemmatization.**  To reduce word variations to their base forms, we apply lemmatization using the `WordNetLemmatizer` from the NLTK library (Bird & Loper, 2004). Part-of-speech (POS) tagging is employed to accurately identify the correct lemma for each word. For example, '`running`' (verb) is converted to '`run`', while '`games`' (noun) is converted to '`game`'.

**Whitespace Normalization.**  Finally, multiple consecutive spaces and newlines are replaced with a single space to maintain text cleanliness.

**Illustrative Example.**  The following text demonstrates the preprocessing pipeline:

- **Original Text:** `"I can't believe it! Visit our website at https://example.com. Check out our 24-hour service and wi-fi options!"`
- **Processed for BM25 Retrieval:** `"cannot believe visit our website check out our twentyfourhour service wifi options"`

**Preserving Original Sentences for Synthesis.**  It is important to note that the preprocessed text is only used to calculate relevance scores during the **BM25 retrieval** stage. After ranking the sentences based on relevance, the original, unprocessed sentences are retained and passed to the **Synthesizer stage** for input to the LLMs. This approach ensures that the LLMs receive high-quality, coherent inputs for generating accurate and structured summaries.

This preprocessing pipeline is essential for enhancing lexical matching between user queries and review sentences, but is not applied for **Dense retrieval**, which directly leverages the original sentences to capture semantic relationships.

# E   Case Study: Analysis of AOS Verification Metrics

This section presents illustrative challenging examples identified during manual inspection of the generated outputs. For simplicity, we showcase a single piece of evidence to illustrate various cases.

## E.1   Aspect Relevance (AR)

**Description:** This metric evaluates whether the most frequently mentioned aspect in the retrieved evidence aligns with the generated highlight. Challenges arise from:

---

[11] https://github.com/kootenpv/contractions

- **Implicit vs. Explicit Aspect Mismatch:** For instance, *"ocean view"* may be related to the broader category of *"location"*.
- **Aspect Generalization:** Broad aspects like *"Room amenities"* vs. specific aspects like *"Wi-Fi availability"*.

---

**Aspect Relevance (AR) Examples**

- **Implicit vs. Explicit Aspect:** *Evidence:* "Amazing ocean views from the balcony."
  *Generated Highlight:* "The location offers beautiful views."
- **Aspect Generalization:** *Evidence:* "Limited gym facilities and amenities."
  *Generated Highlight:* "Other amenities were excellent, except gym."

---

### E.2 Sentiment Factuality (SF)

**Description:** This metric checks whether the sentiment polarity of the generated highlight matches the predominant sentiment from the evidence. Challenges include:

- **Positive Bias:** LLMs may downplay negative sentiments.
- **Mixed Sentiment Overgeneralization:** Combining conflicting sentiments into a positive or neutral summary.

---

**Sentiment Factuality (SF) Examples**

- **Downplayed Negative Sentiment:** *Evidence:* "The Wi-Fi was absolutely terrible."
  *Generated Highlight:* "The Wi-Fi was a bit slow."
- **Overgeneralization of Mixed Sentiment:** *Evidence:* "The room was spacious but noisy."
  *Generated Highlight:* "The room was spacious and comfortable."
- **Positive Overstatement:** *Evidence:* "The reception was average."
  *Generated Highlight:* "The reception was excellent."

---

### E.3 Opinion Faithfulness (OF)

**Description:** This metric verifies how well the opinion in the generated highlight aligns with those extracted from the retrieved evidence. Challenges include:

- **Opinion Overstatement:** Overly positive or exaggerated language compared to the original.
- **Semantic Drift:** Divergence due to the rich vocabulary of LLMs.

---

**Opinion Faithfulness (OF) Examples**

- **Overstatement of Opinion:** *Evidence:* "The beds were comfortable."
  *Generated Highlight:* "The beds were exceptionally luxurious."
- **Semantic Drift:** *Evidence:* "The decor was simple and neat."
  *Generated Highlight:* "The decor was elegant and sophisticated."

---

## F  Baseline Methods

To systematically evaluate the performance, we compare them against several baseline methods, ranging from naive random selection to extractive summarization techniques.

For all extractive baselines, we preprocess customer reviews by tokenizing sentences using the nltk[12] sentence tokenizer Bird & Loper (2004). Additionally, to ensure sentiment alignment, we employ SiEBERT Hartmann et al. (2023), a language model fine-tuned for sentiment classification. SiEBERT has been trained on diverse English datasets, including tweets and reviews[13], making it a robust choice for determining polarity in extracted sentences. For the TextRank Mihalcea & Tarau (2004) and LexRank Erkan & Radev (2004) baselines, we use the implementation from the Sumy library[14].

**Random Baseline.** To establish a naive baseline, we implement a random selection strategy for generating PROS and CONS from user reviews. Given a reference summary for an entity, we first determine the number of PROS and CONS in the summary. Next, we aggregate all candidate sentences from the corresponding customer reviews. We then randomly sample the same number of sentences as the expert PROS. To prevent overlap, these selected PROS are removed before randomly selecting the required number of CONS.

This method does not incorporate any semantic understanding, ranking, or structured summarization, serving as a lower-bound performance estimate. Comparisons with more advanced models highlight their ability to generate structured and meaningful summaries beyond random selection.

**Oracle Baseline.** The oracle baseline selects the most relevant user-generated sentences that best align with the output summary. For each reference PROS and CONS sentence, we retrieve the most similar user review sentence using an extractive matching strategy, where the sentence with the highest ROUGE-L (RL) score is selected as the best candidate.

Additionally, a sentiment constraint is enforced to ensure semantic consistency. A sentiment classifier assigns a polarity label to each candidate sentence, ensuring that extracted PROS align with positive sentiment and CONS with negative sentiment.

This extractive oracle represents an upper-bound performance estimate, as it leverages gold summaries to retrieve the most relevant content from user reviews. Comparing learned models against this baseline assesses their ability to generalize beyond purely extractive methods.

**TextRank Baseline.** TextRank Mihalcea & Tarau (2004) is an unsupervised, graph-based ranking algorithm for extractive summarization. It constructs a similarity graph where sentences serve as nodes, and edges represent lexical similarity. To establish a TextRank-based extractive baseline, we first merge all customer reviews for an entity and extract the top-ranked sentences based on their importance.

To ensure sentiment alignment with expert critiques, we classify each extracted sentence using SiEBERT. The highest-ranked positive sentences are selected as PROS, while the most relevant negative sentences are chosen as CONS, with their respective counts determined by the target reference summary.

This method provides a stronger extractive baseline by integrating both relevance ranking and sentiment filtering. However, it remains limited to surface-level extraction without generating novel insights or restructuring content.

**LexRank Baseline.** LexRank Erkan & Radev (2004) is an unsupervised extractive summarization algorithm that ranks sentences based on their centrality within a document. It constructs a similarity graph where each sentence is a node, and edges represent cosine similarity between sentence embeddings. Using a modified PageRank algorithm, it assigns higher importance to sentences that are representative of the overall document.

For the LexRank-based extractive baseline, we first aggregate customer reviews for an entity and apply LexRank ranking. To ensure alignment with the target summary, we classify

---

[12] https://www.nltk.org/

[13] sentiment-roberta-large-english

[14] https://github.com/miso-belica/sumy

each sentence's sentiment using `SiEBERT` and select the highest-ranked positive sentences as PROS and negative sentences as CONS, ensuring that the selected sentences match the structure of the target summary.

This approach ensures that extracted summary reflect both centrality and sentiment alignment, but it remains limited to extractive selection, lacking the ability to generate structured insights or rephrase user content.

# G   Extended Discussion of Related Work

## G.1   Opinion Summarization Methods

Opinion summarization has been extensively studied under extractive and abstractive paradigms. Extractive methods generate summaries by directly selecting representative sentences from user reviews (Nayeem & Chali, 2017b; Angelidis et al., 2021; Basu Roy Chowdhury et al., 2022; Li et al., 2023; Chowdhury et al., 2024; Li & Chaturvedi, 2024). While these approaches ensure factual consistency by preserving source content, they often produce verbose and disjointed summaries that lack coherence. Abstractive methods, on the other hand, synthesize content by paraphrasing and restructuring input text (Ganesan et al., 2010; Nayeem, 2017; Chali et al., 2017; Nayeem & Chali, 2017a; Nayeem et al., 2018; 2019; Chu & Liu, 2019; Fuad et al., 2019; Bražinskas et al., 2020b; Amplayo & Lapata, 2020; Hosking et al., 2024b), offering greater fluency and readability. However, such methods are prone to hallucinations and factual inconsistencies.

The advent of LLMs has significantly influenced opinion summarization. Recent works have explored using LLMs to generate summaries even in zero-shot or few-shot settings (Bhaskar et al., 2023; Siledar et al., 2024a). However, existing research primarily focuses on short-form inputs or generating generic, paragraph-style summaries, which fail to address the challenges posed by long-form user feedback where the input length can exceed 100K tokens (Chang et al., 2024). Moreover, current approaches lack mechanisms for structured, query-specific summarization, limiting their applicability for user-centric tasks.

To the best of our knowledge, `OpinioRAG` is the first framework specifically designed for user-centric summarization of large-scale, long-form user reviews. Our approach employs a RAG-based paradigm that combines retrieval and generation, enabling efficient processing of extensive review corpora. Unlike prior methods, we generate structured summaries segmented into PROS and CONS presented as key-point-style highlights, improving both readability and usability. This structured format allows users to quickly access relevant insights based on their specific queries, which is particularly beneficial for real-world decision-making (Appendix Section C).

## G.2   Benchmarks for Opinion Summarization

Creating large-scale annotated datasets for opinion summarization remains a major challenge. Unlike news summarization, where summaries are often provided within the source documents (Hermann et al., 2015; See et al., 2017; Narayan et al., 2018), user review platforms rarely offer structured summaries. Generating high-quality summaries through human annotation is costly and time-consuming. Consequently, prior studies have utilized self-supervised methods by treating individual reviews as pseudo-summaries while using the remaining reviews as input (Amplayo & Lapata, 2020; Elsahar et al., 2021). However, these datasets are generally limited to fewer than 10 reviews per entity (Angelidis & Lapata, 2018; Chu & Liu, 2019; Bražinskas et al., 2020a), with only a few extending to hundreds (Angelidis et al., 2021; Bražinskas et al., 2021).

Most existing datasets focus on short or mid-length inputs, with average input lengths significantly below 100K tokens (Chang et al., 2024). In contrast, our `OpinioBank` dataset is the first large-scale dataset featuring long-form user reviews exceeding thousands of reviews per entity, paired with unbiased expert reviews and manually annotated queries. This comprehensive setup enables the evaluation of LLMs under realistic, high-volume review scenarios, where existing methods fall short.

### G.3 Query-focused Summarization and RAG

Query-focused summarization (QFS) aims to generate summaries tailored to specific queries, enhancing personalization and control (Xu & Lapata, 2020; Vig et al., 2022). The QFS concept has been extended to opinion summarization, particularly in aspect-controllable settings, where summaries are generated based on pre-defined aspects. Existing datasets, such as OpoSum+ (Amplayo et al., 2021) and SPACE (Angelidis et al., 2021), focus on general aspects common across all entities (e.g., "location," "service," "food") derived from short-form user reviews. In contrast, OpinioBank offers a diverse range of entity-specific queries tailored to individual entities and derived from long-form reviews. These queries include detailed aspects such as "tuk tuk service," "museum access," and "yoga classes," providing a richer representation of user needs and preferences.

Previous RAG-based approaches, like (Hosking et al., 2024b), utilize hierarchical clustering for coherent summaries by retrieving sentence clusters based on encoded paths rather than direct retrieval methods. However, this approach lacks scalability and mechanisms for estimating factual alignment, making it unsuitable for handling very long inputs. In contrast, OpinioRAG leverages a RAG-based framework designed for scalability, controllability, and verifiability. It allows integration of structured verification metrics—Aspect Relevance (AR), Sentiment Factuality (SF), and Opinion Faithfulness (OF)—to estimate factual alignment and interpretability. Unlike clustering-based methods, our modular design enables the integration of various retrievers and LLMs, making OpinioRAG adaptable and effective for generating accurate, query-specific summaries.

Although RAG has been successfully applied to domains such as *financial* (Wang et al., 2025), *legal* (Pipitone & Alami, 2024), dialogue response generation from subjective knowledge (Zhao et al., 2023), news headline generation (Shohan et al., 2024), and general knowledge retrieval across *economics*, *psychology*, *mathematics*, and *coding* (SU et al., 2025), to the best of our knowledge, OpinioRAG is the first framework to apply RAG for generating query-specific, structured summaries from extensive, noisy, long-form user reviews.

## H Limitations

In this work, we evaluated our proposed framework using a selection of both open-source and closed-source LLMs, prioritizing cost-effective models that can be deployed on consumer-grade hardware, given the constraints of *academic settings*. The performance of more powerful, large-scale models remains unexplored. We encourage the broader research community to benchmark such models using our dataset and methods.

While we experimented with different retrievers (BM25 and Dense) using Top-K values of 5 and 10, optimizing for other retrieval configurations is beyond the scope of this study. We acknowledge that exploring additional retrievers or hybrid retrieval strategies may enhance performance.

Our proposed verification metrics, built on the RAG framework, enable scalable assessment by processing manageable evidence units. However, their applicability may be limited when applied to full long-form settings without segmented retrieval. Future work could address these limitations by adapting the metrics for end-to-end, long-form input processing.

We focus on the hotel domain, which proves advantageous due to the availability of long-form user reviews and corresponding expert summaries. While our primary focus is on hotel reviews, our training-free framework is designed to be generalizable across various domains, such as product reviews or forum discussions, provided that reference summaries are available for evaluation. Although fine-tuning models could enhance performance, we intentionally demonstrate a training-free approach to avoid reliance on large labeled datasets for supervised training. However, we acknowledge that fine-tuned specialist models could further improve quality.

Human evaluation of the generated summary is challenging in long-form settings. We did not perform manual human evaluation. However, we propose verification metrics under our RAG framework and provide a case study with some manual inspection of

---

**LLM-as-a-Judge Evaluation Rubric**

**1. Aspect Relevance (AR):**
*Does the system summary cover the same topics or facets as the expert summary (even if worded differently)?*

 5:  Nearly all aspects are clearly covered or rephrased accurately.
 4:  Most key aspects appear present with only minor omissions.
 3:  Some aspects included but important ones missing.
 2:  Limited overlap; several relevant topics are absent.
 1:  Very few expert aspects are represented.

**2. Non-Redundancy (NR):**
*Are aspects mentioned only once? Are key points repeated or paraphrased redundantly?*

 5:  No redundancy at all.
 4:  Rare repetition of content or phrasing.
 3:  Occasional redundancy but not disruptive.
 2:  Multiple repeated points or phrases.
 1:  Severe redundancy across the summary.

**3. Sentiment Agreement (SA):**
*Is the tone (positive or negative) about aspects consistent between the summaries?*

 5:  Sentiment matches almost perfectly across shared aspects.
 4:  Sentiment is aligned with minor acceptable tone shifts.
 3:  Mixed polarity agreement; some aspects differ noticeably.
 2:  Sentiment regularly conflicts or distorts meaning.
 1:  System contradicts or inverts sentiment frequently.

**4. Opinion Faithfulness (OF):**
*Are the factual or evaluative claims in the system summary grounded in the expert summary?*

 5:  All claims are consistent or sensible reformulations.
 4:  Minor creative liberties but still factually plausible.
 3:  Some claims deviate or are weakly grounded.
 2:  Frequent discrepancies or unsupported elaborations.
 1:  Mostly fabricated or misleading content.

**5. Overall Usefulness (OU):**
*Would the system summary help a potential customer make a reasonable decision?*

 5:  Equally or more informative than the expert version.
 4:  Informative with small shortcomings.
 3:  Provides basic guidance but lacks confidence.
 2:  Weak or unclear assistance.
 1:  Confusing or unhelpful overall.

Figure 4: LLM-as-a-Judge evaluation rubric used to assess the quality of summaries along five dimensions: aspect relevance, redundancy, sentiment alignment, opinion faithfulness, and overall usefulness.

the generated outputs in Appendix E. We present some examples of output summaries in Appendix I and conduct an *"LLM-as-a-Judge"* evaluation (Section 5)—a scalable and cost-effective alternative to traditional human evaluation, which is increasingly adopted nowadays in recent works (Gu et al., 2025).

Lastly, our research and the development of OpinioRAG are focused exclusively on the **English language**. Its applicability and effectiveness for other languages remain unexplored and warrant future investigation.

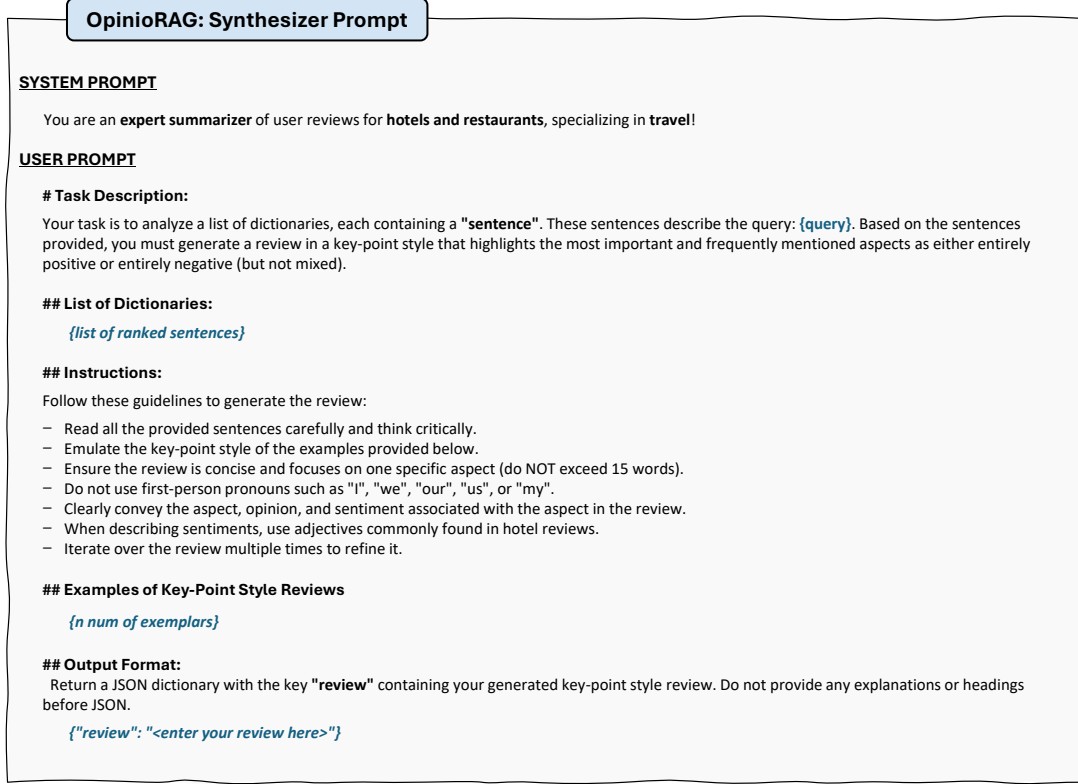

Figure 5: The prompt for our `OpinioRAG` - Synthesizer Stage.

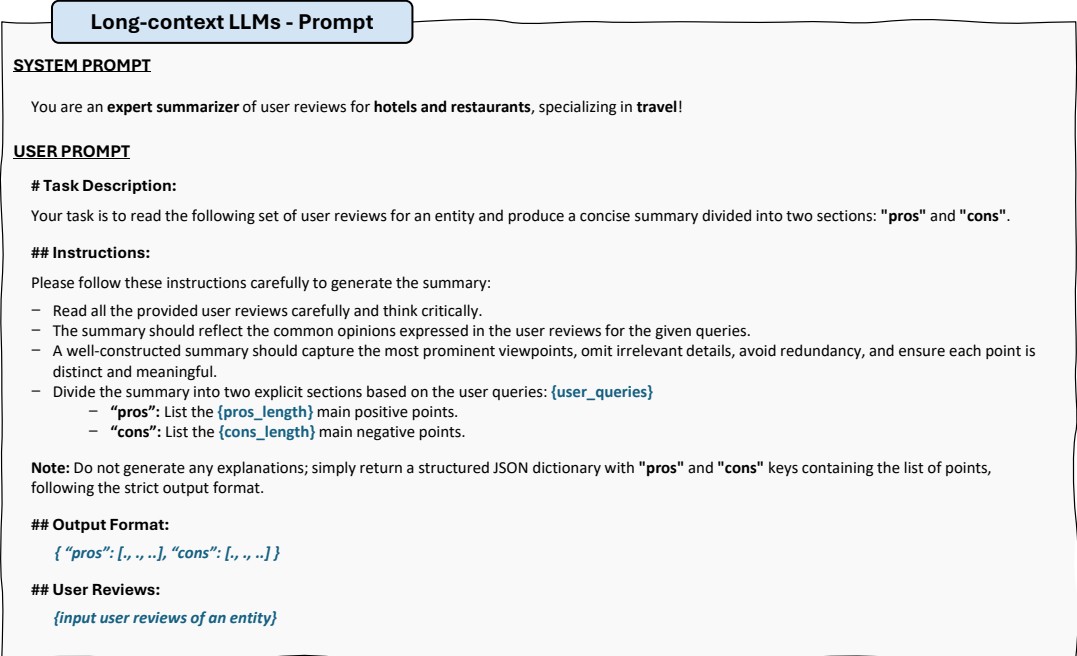

Figure 6: The prompt for Long-context LLMs baselines.

# I  Example Output Summaries

We present generated summaries from our OpinioRAG framework alongside a reference summary to facilitate head-to-head comparisons across different settings and baseline Long-

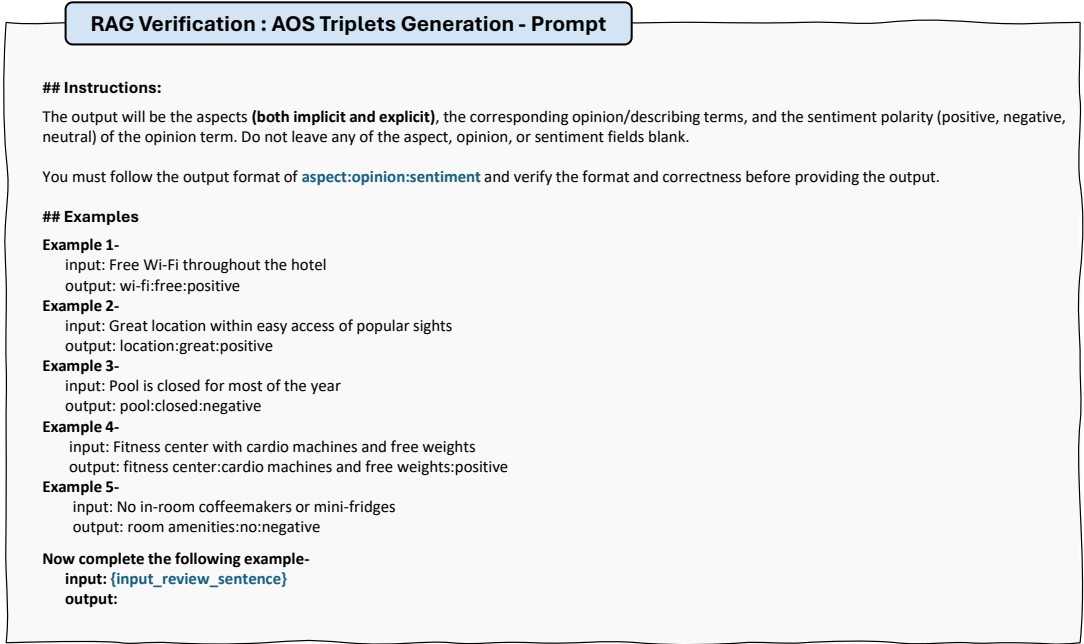

**RAG Verification : AOS Triplets Generation - Prompt**

## Instructions:

The output will be the aspects **(both implicit and explicit)**, the corresponding opinion/describing terms, and the sentiment polarity (positive, negative, neutral) of the opinion term. Do not leave any of the aspect, opinion, or sentiment fields blank.

You must follow the output format of **aspect:opinion:sentiment** and verify the format and correctness before providing the output.

## Examples

**Example 1-**
   input: Free Wi-Fi throughout the hotel
   output: wi-fi:free:positive
**Example 2-**
   input: Great location within easy access of popular sights
   output: location:great:positive
**Example 3-**
   input: Pool is closed for most of the year
   output: pool:closed:negative
**Example 4-**
   input: Fitness center with cardio machines and free weights
   output: fitness center:cardio machines and free weights:positive
**Example 5-**
   input: No in-room coffeemakers or mini-fridges
   output: room amenities:no:negative

**Now complete the following example-**
   input: **{input_review_sentence}**
   output:

Figure 7: The prompt for generating AOS triplets for RAG Verification.

context LLMs. The outputs are provided for **Entity ID: 18** and **Entity Name: Polo Towers Suites** from our OpinioBank dataset across all configurations for comprehensive evaluation.

- **OpinioRAG - Open-source Models (BM25 TopK, K=10):** The models evaluated include Llama-3.1-8B, Mistral-7B, and Gemma-2-9B (Table 10).

- **OpinioRAG - Closed-source Models (Dense TopK, K=5):** The models evaluated include GPT-4o-mini, Gemini-2.0-flash, and Claude-3.5-haiku (Table 11).

- **Baseline Long-context Model Outputs:** The models GPT-4o-mini, Gemini-2.0-flash, and Claude-3.5-haiku are evaluated with the given queries (Table 12).

| Models | Pros | Cons |
|---|---|---|
| **Reference Summary** | – Central location, about 200 feet from the Strip
– All rooms have balconies (rare in Las Vegas)
– Rooms have kitchenettes or full kitchens
– Rooftop pool, whirlpool, and barbecue area overlooking the Strip
– On-site convenience store and business center
– On-site gym and hair/nail salon available
– Children's pool and play area
– Free shuttle to grocery store, outlets, and Downtown Las Vegas | – No on-site restaurant for the guests
– Small pool (capacity: 30)
– No poolside bar or drinks service
– Minimal service and limited housekeeping
– Time-share sales representative can come across as pushy
– Daily resort fee (includes Wi-Fi) |
| **Llama-3.1-8B** | – Excellent strip location with easy access to amenities and attractions
– Balconies available in all rooms with great views
– Well-equipped rooms with full kitchens, living areas, and ample space
– Rooftop pool available, often described as awesome and great
– Convenience store and business center available on site
– Full salon, spa, and well-equipped gym available on-site
– Excellent children's areas with pools, splash pads, and play spaces
– Free shuttle service to various locations available | – Limited on-site dining options available
– Pool is often of good size and well-maintained
– No poolside bar available
– Excellent service with daily housekeeping and quick customer service
– Aggressive time-share sales staff with pushy tactics and rude representatives
– Resort fees can be excessive and poorly advertised |
| **Mistral-7B** | – Excellent strip location
– Most rooms feature balconies with scenic views
– Spacious rooms with well-equipped kitchens
– Riveting rooftop pool experience
– On-site business center and convenience store
– Full salon/spa and well-equipped gym
– Well-equipped children's area with pools and play zones
– Convenient shuttle services to local destinations | – No on-site restaurant or bar
– Roof top pools, varying sizes
– Absence of poolside bar
– Excellent and consistent customer service
– Aggressive time-share sales representatives
– Unpredictable resort fees |
| **Gemma-2-9B** | – Excellent location on the strip
– Rooms offer balconies with good views
– Rooms are spacious and feature full kitchens
– Rooftop pool is awesome
– Well-equipped business center available
– Spacious and well-equipped gym available
– Children's area is well-equipped and convenient
– Shuttle service is convenient and extensive | – No on-site restaurant available
– Rooftop pool is clean and a good size
– No on-site bar or restaurant
– Customer service is terrible
– Time-share sales representatives are overwhelmingly aggressive
– Resort fees are often excessive and misleading |

Table 10: **Comparison of Generated Summaries from Open-source models - BM25 TopK (K=10) from our OpinioRAG** for the queries ["strip location", "room balcony", "room kitchen", "rooftop pool", "convenience store", "gym and salon", "child area", "shuttle service", "restaurant", "pool size", "pool bar", "service", "sales representatives", "resort fee"]. The Reference Summary is underlined. The 'Pros' and 'Cons' are highlighted in green and red, respectively. Misaligned generated highlights relative to the expert highlights are marked in blue.

| Models | Pros | Cons |
|---|---|---|
| **Reference Summary** | – Central location, about 200 feet from the Strip
– All rooms have balconies (rare in Las Vegas)
– Rooms have kitchenettes or full kitchens
– Rooftop pool, whirlpool, and barbecue area overlooking the Strip
– On-site convenience store and business center
– On-site gym and hair/nail salon available
– Children's pool and play area
– Free shuttle to grocery store, outlets, and Downtown Las Vegas | – No on-site restaurant for the guests
– Small pool (capacity: 30)
– No poolside bar or drinks service
– Minimal service and limited housekeeping
– Time-share sales representative can come across as pushy
– Daily resort fee (includes Wi-Fi) |
| **GPT-4o-mini** | – Excellent location, just steps away from the strip.
– Rooms feature spacious balconies, enhancing the overall experience.
– Rooms feature fully equipped kitchens, ideal for groups or extended stays.
– Rooftop pool is excellent and highly praised by guests.
– Convenience store and business center conveniently located on-site.
– Well-equipped gym with a full salon and spa services available.
– Excellent children's play area with splash pad and kiddie pool.
– Complimentary shuttle service available to grocery store and other locations. | – No onsite restaurant, but numerous dining options within walking distance.
– Small pool area, but very nice and relaxing.
– Nice pool area with shaded cabanas, but lacks a swim-up bar.
– Outstanding service with friendly and attentive staff
– Aggressive and pushy time-share sales tactics are prevalent.
– High resort fees not clearly disclosed during booking |
| **Gemini-2.0-flash** | – Excellent location, very close and convenient to the strip.
– Rooms feature desirable balconies, some even have two!
– Rooms offer full kitchens, ideal for longer stays and larger groups.
– Rooftop pool is a great amenity.
– Convenience store and business center are convenient amenities.
– Gym and salon are available, with the gym being adequately equipped.
– Excellent child area with splash pad, play area, and kiddie pool
– Convenient shuttle service to airport, grocery store, and other locations. | – No on-site restaurant; however, many dining options conveniently located nearby.
– Pool size is small, but the area is relaxing and nice.
– Pool area is nice but lacks a swim-up bar or any bar.
– Excellent and friendly service consistently impresses guests.
– Aggressive time-share sales representatives; high-pressure tactics are expected.
– Unexpected and overpriced daily resort fee for basic amenities, such as parking. |
| **Claude-3.5-haiku** | – Excellent strip location within walking distance
– Spacious balconies available in multiple rooms
– Convenient in-room kitchen facilities with microwave, fridge, and cutlery
– Impressive rooftop pool with great amenities
– Convenient on-site business center with small convenience store
– Well-equipped gym and full-service salon on-site with multiple amenities
– Excellent child-friendly facilities with play areas and splash zones
– Limited shuttle service with trips to grocery store and airport | – No on-site restaurant, but numerous dining options nearby
– Small but pleasant pool area, relaxing atmosphere
– No pool bar available, despite pleasant pool area with cabanas
– Exceptional and friendly service consistently praised by guests
– Aggressive, pushy time-share sales representatives create unwelcome experience
– Excessive resort fees around 20−28 per day with unclear inclusions |

Table 11: **Comparison of Generated Summaries from Closed-source models - Dense TopK (K=5) from our OpinioRAG** for the queries ["strip location", "room balcony", "room kitchen", "rooftop pool", "convenience store", "gym and salon", "child area", "shuttle service", "restaurant", "pool size", "pool bar", "service", "sales representatives", "resort fee"]. The Reference Summary is underlined. The 'Pros' and 'Cons' are highlighted in green and red, respectively. Misaligned generated highlights relative to the expert highlights are marked in blue.

| Models | Pros | Cons |
|---|---|---|
| **Reference Summary** | – Central location, about 200 feet from the Strip
– All rooms have balconies (rare in Las Vegas)
– Rooms have kitchenettes or full kitchens
– Rooftop pool, whirlpool, and barbecue area overlooking the Strip
– On-site convenience store and business center
– On-site gym and hair/nail salon available
– Children's pool and play area
– Free shuttle to grocery store, outlets, and Downtown Las Vegas | – No on-site restaurant for the guests
– Small pool (capacity: 30)
– No poolside bar or drinks service
– Minimal service and limited housekeeping
– Time-share sales representative can come across as pushy
– Daily resort fee (includes Wi-Fi) |
| **GPT-4o-mini** | – Great location on the Las Vegas Strip, close to attractions, restaurants, and shopping.
– Spacious and clean accommodations with full kitchens and comfortable beds.
– Friendly and helpful staff who provide excellent customer service.
– Rooftop pool with great views and a relaxing atmosphere.
– No casino on-site, providing a quieter environment away from casino noise.
– Free parking options available for guests.
– Daily housekeeping provides fresh towels and clean rooms.
– Access to a convenience store within the hotel for snacks and essentials. | – Aggressive timeshare sales pitches at check-in and throughout the stay.
– Resort fees add to the overall cost and can be perceived as misleading.
– Some guests reported noise from neighboring rooms and maintenance activities.
– Lack of on-site dining options or a bar, requiring guests to go off property for meals.
– Inconsistent housekeeping services, with some guests experiencing lack of thorough cleaning.
– Small rooftop pool may feel crowded during peak times. |
| **Gemini-2.0-flash** | – Excellent location on the strip, close to many attractions, restaurants, and shops.
– Spacious, clean, and well-maintained rooms, some with separate living areas, often described as condo-like.
– Fully equipped kitchens or kitchenettes in the rooms allow for meal preparation and cost savings.
– Friendly and helpful staff providing excellent customer service.
– Rooftop pool and hot tub offer great views and a relaxing experience.
– Free parking and valet service.
– Convenience store on-site for basic needs.
– Quiet atmosphere as it is a non-casino hotel. | – Aggressive timeshare sales representatives can be persistent and detract from the stay.
– Some rooms may be dated or in need of renovation.
– The walls are thin, which causes noise issues between rooms.
– The rooftop pool can be small and crowded.
– Resort fees add to the overall cost.
– Some walkways to the strip can be uninviting, especially at night. |
| **Claude-3.5-haiku** | – Excellent central location on the Las Vegas Strip, close to major attractions.
– Spacious and clean rooms with full kitchens in many suites.
– Rooftop pool with great views of the Strip.
– Quiet environment without a casino, offering a peaceful retreat.
– Friendly and helpful staff.
– Free parking and convenient amenities.
– Walking distance to restaurants, shops, and casinos.
– Well-maintained facilities with fitness center and convenience store. | – Persistent timeshare sales pitches and marketing tactics.
– Small pool size and limited pool facilities.
– Some dated room furnishings and decor.
– Early 10 AM checkout time.
– Additional resort fees not included in initial pricing.
– Occasional noise issues with adjoining rooms.
– Limited housekeeping services in some instances. |

Table 12: **Comparison of Generated Summaries from Long-context LLMs baselines** for the queries ["strip location", "room balcony", "room kitchen", "rooftop pool", "convenience store", "gym and salon", "child area", "shuttle service", "restaurant", "pool size", "pool bar", "service", "sales representatives", "resort fee"]. The Reference Summary is underlined. The 'Pros' and 'Cons' are highlighted in green and red, respectively.

