# OpenReview forum: "OpinioRAG: Towards Generating User-Centric Opinion Highlights from Large-scale Online Reviews"
_colmweb.org/COLM/2025/Conference — COLM 2025_

### Official Review · Reviewer_Pz2B · 2025-05-11

**Rating:** 6
**Confidence:** 4
**Ethics Flag:** 1

**Summary:**

This paper introduces OpinionRAG, an RAG framework for summarizing online reviews using LLMs with extra verifications. It also creates OpinioBank: a new hotel review dataset with thousands of reviews per hotel, paired with existing online summaries and manually annotated user queries. Results show that BM25 outperforms dense retrieval, and open-source LLMs outperform closed-source LLMs on this task based on automatic metrics such as ROUGE.

**Reasons To Accept:**

- The newly collected dataset can be potentially interesting for opinion summarization research.

- The paper is well-presented, with clear figures and tables highlighting the key characteristics of the new dataset, the proposed approach, and the main results.

**Reasons To Reject:**

- It’s not clear about the main contribution of this work.
  - RAG has been widely used for summarization, even for review summarization [1,2], and seems the main technical contribution of this work is not RAG for summarization, but a further verification step that can be applied to any RAG approaches. Since it's not shown in Fig 1, I regard this step more as a post-hoc evaluation that hasn’t been integrated to summarization.
  - If so, the paper's focus should shift to a meta-evaluation of this verification method, such as comparisons against LLM auto-raters and human judgments. Given this, the rationale for collecting a new dataset with paired summaries is also less compelling as the verification is "reference-free."

- The proposed three verification steps sound not reasonable to me. As I mentioned above, meta-evaluation is needed.
  - Aspect Relevance: this step relies on a pre-trained Aspect-Opinion-Sentiment triplet extractor trained on SemEval data. I think this data is for laptop and restaurant reviews and the model trained on this data is hard to apply to cross-domain. Not clear to me why it can extract aspects from hotel reviews. Also, since it compares aspects between the summary and the retrieved evidence, it cannot detect irrelevance when the retrieved evidence is irrelevant to the user query.
  - Sentiment Factuality: Using majority sentiment as an indicator for sentiment factuality seems overly simplistic, as online review distributions are often skewed to positive. Say, a hotel with 51% positive sentiment versus one with 99% positive sentiment should have different sentiment in their summaries.
  - Opinion Faithfulness: This considers the cosine similarity between a generated opinion and any single retrieved opinion, which relies heavily on the quality of the underlying opinion representations. It is known that general-purpose semantic embeddings often struggle to capture the semantic relationships in opinion expressions. For example, embeddings of "large" and "small" might be more similar than "large" and "spacious". The paper didn’t specify which embedding were used and if the chosen embedding is adequate.

- Dataset quality
  - I’m concerned about the method of collecting reviews from one platform and summaries from another. Reviews are dynamic and reflect the state of the establishment at a particular time. It is not guaranteed that summaries from one source, potentially written at a different time or based on a different underlying review pool, will faithfully represent reviews collected elsewhere.
  - The paper mentioned that 1,456 queries were manually annotated to "ensure query diversity and minimize redundancy", but it’s not mentioned how this is ensured and minimized.

- Lack of evaluating retrieval module
  - The paper lacks evaluation of the retrieval module such as its precision and recall. This is important as errors during retrieval will inevitably propagate to the generated summaries and cause misleading outputs. For example, when negative reviews are prevalent while top-k contains more positive reviews, the summary will be wrongly positive.
  - In addition, the selling-point of this paper is the ability to handle large-scale review sets. But if the system ultimately relies on only a small top-k set (say, 10) instead of ALL query-relevant reviews for summarization, I don’t see the benefit of scaling the review pool to thousands per entity.

- Results and conclusions are not convincing
  - The authors acknowledge the known limitations of ROUGE and BERTScore for evaluating summarization tasks, yet still draw conclusions from these scores. More reliable auto metrics and human evaluation are required for evaluation.
  - The finding that all evaluated open-source models outperform closed-source models sounds suspicious and runs contrary to many observations in other research. Again, this raises questions about the reliability of the dataset and the evaluation metrics.

[1] What do others think? Task-Oriented Conversational Modeling with Subjective Knowledge

[2] Hierarchical indexing for retrieval-augmented opinion summarization

---

> ### Author Response · Authors · 2025-06-02
> **Authors' Response (Part 1/3)**
>
> We thank the reviewer for raising important points. We respond to each below:
>
> ## **Main contribution of this work and RAG**
>
> While RAG-based generation has been explored in prior work, the key contribution of our paper lies in **formulating and benchmarking user-centric summarization** from long-form, noisy, and diverse user reviews. Specifically:
>
> 1. We introduce **OpinioBank**, a **novel dataset** pairing long-form user reviews (∼1,000 per entity) with expert-written structured summaries (PROS/CONS) and manually curated queries to guide retrieval.
>
> 2.  We propose a **training-free, modular RAG framework** tailored for this real-world setting, where retrieval is query-guided and operates over large, heterogeneous opinion corpora (**Table 1**).
>
> 3. We introduce a **verification module**, applied only to the Synthesizer stage, that uses Aspect–Opinion–Sentiment (AOS) triplets to assess factual grounding and sentiment consistency with the retrieved user reviews.
>
> -----------------------
>
> ## **Regarding two cited works**
>
> - **“What Do Others Think?”** targets task-oriented dialogues, retrieving subjective evidence to fill slots. In contrast, our task is structured opinion summarization from extensive user review corpora—a fundamentally different objective.
> - **(Hosking et al., 2024b)**, which we **already discuss in the paper**, retrieves clustered review sentences via encoded paths. Our approach, by contrast, uses explicit user queries and decouples retrieval from generation, mirroring the standard RAG framework.
>
> For broader positioning, we provide an extended discussion of RAG-based methods in the Related Work section (`Lines 1099–1113`). We hope this clarifies that while our work builds on RAG, the core innovations lie in the dataset, framework design, and structured verification.
>
> -----------------------
>
>
> ## **Dataset Quality**
>
> **Pairing Across Platforms:** Although user reviews and expert summaries are sourced from different platforms, they are paired via unique entity identifiers. Crucially, our dataset construction pipeline includes an **explicit verification module** (`Lines 130–135`): we use annotated **query terms as anchors** to verify the presence of supporting evidence in user reviews. If no relevant evidence is found, the corresponding query is **excluded from the gold summary**, ensuring only grounded content remains.
>
> **Empirical Validation:** To assess the reliability of this pairing, we compare OpinioBank against fully human-annotated datasets (Appendix A2, `Lines 784–790`). As shown in **Table 6**, OpinioBank achieves **higher alignment with source reviews** than existing human-annotated datasets, demonstrating that our verification protocol is both effective and principled.
>
> **Temporal Dynamics:** We agree that review content is time-sensitive. However, our setting involves long-form review corpora, which enables models to detect recurrent themes and infer stable consensus over time. Furthermore, OpinioBank provides review metadata such as posting date and helpful votes (`Line 775`), which can be used to filter reviews temporally. We discuss how these metadata can support **modeling temporal dynamics** in Section 6 (`Lines 320–329`).
>
> **Query Diversity and Redundancy:** We annotated **5,975 queries** across the entities (Table 1), with 1,456 unique queries detailed in **Table 5**. To minimize redundancy and maximize diversity (`Lines 124–129`), we systematically generalize or specialize queries within an entity. For example, if *“room”* is used once, subsequent annotations may use more specific forms like *“room coffeemakers”* or *“room lighting”*.
>
> -----------------------
>
> ## **Evaluation of Retrieval Module**
>
> We primarily use retrieval to extract relevant snippets from long-form, large-scale, noisy user reviews and eliminate clutter. However, we extensively evaluate its effect on final summary quality across retrieval types, Top-K settings, and through ablations. In our **ablation study** (`Lines 252–255`), we use only the top-most retrieved sentence as the summary—without the Synthesizer—and present the results in **Table 2**, discussed in `Lines 259–261`.
>
> Regarding Top-K, we show that increasing retrieval size from Top-5 to Top-10 consistently improves performance across models (see **Table 7** in the Appendix for Top-5 results), highlighting the benefit of capturing dispersed evidence (`Lines 263–265`). We also acknowledge this in `Lines 1121–1124`, suggesting that larger K values could further enhance performance.

---

> > ### Author Response · Authors · 2025-06-02
> > **Authors' Response (Part 2/3)**
> >
> > ## **Proposed Verification Methods**
> >
> > We position these components as **“verification”** methods (`not evaluation`) throughout the paper. Their primary purpose is to assess whether the LLM-generated highlights from the Synthesizer stage are **grounded in the retrieved user review snippets**. Our RAG framework is explicitly decoupled into retrieval and synthesis, which enables this targeted verification. As explicitly discussed in `Lines 199–205`, verification is intended to check whether specific grounding requirements are met, whereas **“evaluation”** assesses overall quality and typically requires meta-evaluation or human judgments.
> >
> > Additionally, we conducted a **case study** through manual inspection to analyze verification behavior (**Appendix E**, `Lines 960–985`). The verification module is reference-free and modular (`Lines 205–208`), enabling the use of open-source models such as **Scaria et al. (2024)** in *academic settings*. Our modular design is future-proof and supports integration of **stronger triplet extractors** as they emerge; we also share our prompting template in **Figure 5**.
> >
> > Now, addressing the individual components:
> >
> > - **Aspect Relevance:** OpinioBank includes reviews from hotels, resorts, and restaurants, as shown in **Figure 2**. Given the high similarity between hotel and restaurant domains, the triplet extractor we use (Scaria et al., 2024) is suitable—`Table 8` in [their paper shows](https://aclanthology.org/2024.naacl-short.63/) strong cross-domain performance when trained on `Rest15` and tested on `Hotel15`, outperforming other settings due to domain closeness.
> >
> > - **Sentiment Factuality:** As discussed in Section 3.3, our goal is to verify whether the Synthesizer output aligns with the retrieved evidence. We also discuss sentiment distribution considerations throughout the evaluation, further analysis, and future directions sections.
> >
> > - **Opinion Faithfulness:** We use a pre-trained model fine-tuned on restaurant reviews (https://huggingface.co/saitejautpala/bert-base-yelp-reviews) to obtain embeddings and compute cosine similarity (*we will include this in the revised paper—thank you for the correction*). Importantly, cases like *“large”* vs. *“small”* **will not arise in our verification module**, as opinion faithfulness is measured in a structured way—given an aspect and its associated sentiment. For details, please refer to `Lines 236–243`.
> >
> > -----------------------
> >
> > ## **Evaluation and LLM-as-a-Judge**
> >
> > We report ROUGE and BERTScore to ensure compatibility with recent studies on opinion summarization, to benchmark our OpinioBank dataset, and to contribute to the discussion on enhancing automatic evaluation metrics (`Lines 818–823`). Based on your comment (*also echoed by Reviewer T7ko*), we conducted a comprehensive **LLM-as-a-Judge evaluation** using GPT-4o across all samples in our OpinioBank dataset. We assessed the best-performing open-source and closed-source models under the OpinioRAG framework (**Table 2**), spanning multiple retrieval strategies and TopK configurations. Each summary was evaluated on a **5-point Likert scale** across the following quality dimensions.
> >
> > As shown in the table below, the `Claude-3.5-haiku + Dense + Top-10` configuration consistently outperforms others, particularly in factual grounding and pragmatic coherence—**highlighting the advantage of closed-source models**. We hope this evaluation reinforces our contributions, and we appreciate your valuable suggestion.
> >
> >
> > **Evaluation Criteria**
> >
> > 1. **Aspect Relevance (AR):** Does the system summary cover the same topics or facets as the expert summary (even if worded differently)?
> > 2. **Non-Redundancy (NR):** Are aspects mentioned only once? Are key points repeated or paraphrased redundantly?
> > 3. **Sentiment Agreement (SA):** Is the tone (positive or negative) about aspects consistent between the summaries?
> > 4. **Opinion Faithfulness (OF):** Are the factual or evaluative claims in the system summary grounded in the expert summary?
> > 5. **Overall Usefulness (OU):** Would the system summary help a potential customer make a reasonable decision?
> >
> > | Model Name       | Retrieval | TopK |  AR   |  NR   |  SA   |  OF   |  OU   |
> > |------------------|-----------|------|-------|-------|-------|-------|-------|
> > | Llama-3.1-8B      | BM25      | 5    | 3.17  | 3.83  | 2.92  | 2.82  | 3.12  |
> > |                  |           | 10   | 3.26  | 3.81  | 2.99  | 2.87  | 3.20  |
> > |                  | Dense     | 5    | 3.23  | **3.94** | 3.04  | 2.89  | 3.19  |
> > |                  |           | 10   | 3.31  | 3.84  | 3.10  | 2.93  | 3.26  |
> > | Claude-3.5-haiku | BM25      | 5    | 3.20  | 3.82  | 2.94  | 2.90  | 3.17  |
> > |                  |           | 10   | 3.25  | 3.73  | 3.02  | 2.95  | 3.22  |
> > |                  | Dense     | 5    | 3.28  | 3.85  | 3.08  | 2.95  | 3.23  |
> > |                  |           | 10   | **3.38** | 3.73  | **3.13** | **2.98** | **3.33** |

---

> > ### Author Response · Authors · 2025-06-02
> > **Authors' Response (Part 3/3)**
> >
> > ## **Open-source vs. Closed-source**
> >
> > In the RAG Verification Assessment (**Table 3**), closed-source models actually outperformed open-source models, indicating that **closed-source synthesizers are more effectively grounded to the retrieved evidence (i.e., *user reviews*)**. Similarly, in the *new LLM-as-a-Judge evaluation* described above, *closed-source models again demonstrated superior performance*.
> >
> > Additionally, our proposed modular framework (OpinioRAG) significantly simplify the task—**an advantage also noted by Reviewer T7ko**—by decoupling it into retrieval and synthesis stages. This design *likely contributed to making open-source models more competitive within our framework*.
> >
> > We also present results for the full end-to-end long-context (without OpinioRAG) setting in **Table 2**, as described in `Lines 838–844`, with the corresponding prompt shown in **Figure 4**. To further strengthen our argument and answer your question, we additionally evaluated **two long-context open-source models** in a full end-to-end setting, where all user reviews are concatenated and passed as input. The results, shown in the following table, clearly demonstrate that open-source models clearly lag behind when operating **under such extreme input conditions**.
> >
> >
> > | Long-context LLMs   | CL   | R1 (PROS) | R2 (PROS) | RL (PROS) | BS (PROS) | R1 (CONS) | R2 (CONS) | RL (CONS) | BS (CONS) |
> > |---------------------|------|-----------|-----------|-----------|-----------|-----------|-----------|-----------|-----------|
> > | Phi-3.5-3.8B        | 128K | 26.67     | 4.89      | 15.52     | 62.45     | 16.75     | 2.37      | 11.07     | 56.74     |
> > | Llama-3.1-8B        | 128K | 27.31     | 5.14      | 16.05     | 62.93     | 17.39     | 2.54      | 11.28     | 57.16     |
> > | GPT-4o-mini         | 128K | 29.97     | 5.76      | 17.21     | 64.95     | 18.97     | 2.70      | 12.22     | 59.94     |
> > | Claude-3.5-haiku    | 128K | **32.70** | **7.03**  | **19.30** | **67.37** | 20.07     | 3.03      | 13.44     | **61.41** |
> > | Gemini-2.0-flash    | 1M   | 30.62     | 5.75      | 17.87     | 65.45     | **20.81** | **3.73**  | **13.70** | 60.92     |
> >
> > -----------------------
> >
> > We believe this comprehensive rebuttal **directly addresses all of your concerns**. We sincerely thank you for your thoughtful and constructive feedback, which has significantly helped us improve the clarity and scope of our work. If there are any remaining concerns or further questions, we would be happy to address them.

---

> > > ### Author Response · Authors · 2025-06-06
> > > **Follow-Up from Authors**
> > >
> > > Dear Reviewer Pz2B,
> > >
> > > Thank you again for your thoughtful and detailed review. We wanted to respectfully follow up and confirm that we have **fully addressed all of your major concerns** in our response.
> > >
> > > We hope our clarifications have conveyed the value and rigor of our work. We would sincerely appreciate it if you would consider **revisiting your score**. Please don’t hesitate to let us know if any questions remain.
> > >
> > > Sincerely,
> > >
> > > The Authors

---

> > > > ### Comment · Reviewer_Pz2B · 2025-06-07
> > > > **Thank you for the rebuttal**
> > > >
> > > > Thank you for the rebuttal. I guess my original concerns were not very clear. Allow me to make them more direct:
> > > >
> > > > 1.  ROUGE is not a reliable metric for opinion summarization. For example, "This hotel is good" and "This hotel is bad" could have a high rouge score despite their opposite meanings. The rebuttal repeatedly refer to rouge results to demonstrate the quality of the data and the effectiveness of the algorithm, which is not convincing.
> > > >
> > > > 2.  “What Do Others Think?” proposes a RAG method for summarizing reviews based on user queries in a dialogue format. It's not about slot filling and is not fundamentally different from your goal. By the way, what is your interface for users to select a query?
> > > >
> > > > 3.  It's not clear to me how verification impacts the summary. The rebuttal says it's applied to the Synthesizer Stage, but I didn't find anything in Section 3.2 (Synthesizer Stage). Are summary bullet points that fail verification simply removed? This needs to be explicitly detailed.
> > > >
> > > > 4.  Thank you for adding the LLM-as-a-Judge evaluation. The new evaluation results now contradict the initial ones, which may reshape the paper's conclusions. A more fundamental question is, given an evaluation result (either rouge or LLM judge), how do we know if it's reliable? That's why I keep recommending human evaluation.
> > > >
> > > > 5.  In applied research like review summarization, when we propose an algorithm, we expect to see a quantitative analysis to verify its efficiency. Therefore, I'd like to know the accuracy of the verifier and the precision and recall of the retriever. For the verifier, the rebuttal only mentioned a case study with negative cases, which actually suggests these verifiers are not working as expected. For the retriever, I understand there are some indirect ROUGE scores, but rouge is not convincing and these results don't answer my question either.
> > > >
> > > > 6.  I don't think the verifiers make sense for the reasons I listed above: 1) The aspect extractor cannot be applied to domains dissimilar to restaurant. 2) Using a majority for contradictory sentiments is oversimplified. 3) Embeddings may not capture the similarity of opinions. The paper uses “beautiful” and “stunning” as an example; I just used another example of "large/spacious/small." I don't understand why the authors think it's not an issue.
> > > >
> > > > Since most of my concerns actually haven't been addressed, I'll keep the score as it is.

---

> > ### Author Response · Authors · 2025-06-09
> > **Final Clarifications and Rebuttal from the Authors**
> >
> > Thank you for your thoughtful follow-up. We appreciate the opportunity to provide further clarification of your direct questions.
> >
> > 1. We agree that ROUGE has limitations; that’s why we also report **BERTScore**, an embedding-based metric. Following established practices in opinion summarization, we include both metrics to ensure compatibility with prior recent works (Bhaskar et al., 2023; Lei et al., 2024; Siledar et al., 2024b; Hosking et al., 2024b), benchmark our OpinioBank dataset (`Lines 815–826`), and contribute to ongoing discussions on automatic metrics (`Section 6`).
> >
> > 2. The original RAG (Lewis et al., 2020) has been **widely applied to many tasks**. Our OpinioRAG builds on it to benchmark structured opinion summarization on OpinioBank. The *“What Do Others Think?”* paper is one such application of RAG for dialogue response generation from subjective knowledge. We discuss other RAG adaptations (e.g., in financial, legal, coding domains) in our Related Work section and **will cite the paper you mentioned** in the final version. Regarding your question—*“what is your interface for users to select a query?”*—queries are manually annotated from expert summaries and included in the OpinioBank dataset (`Lines 120–129`).
> >
> > 3. The verification module’s core purpose is to **assess factuality** of generated highlights given a retrieved evidence (**Section 3.3** and `Lines 196–197` ). It verifies how well the **Synthesizer** (i.e., the LLM) grounds its generation in the retrieved user review snippets. It is **not used** within the summarization pipeline (`Figure 1`), but rather as an external factuality checker, tailored for sentiment-rich domains. The experimental findings are detailed in **Section 4.2**. We agree it could be clearer and are committed to making it more explicit in the revised version.
> >
> > 4. The LLM-as-a-Judge results shared in the **previous response** are for the best-performing open-source and closed-source models under the OpinioRAG framework; their ROUGE and BERTScore scores are already comparable (**Table 2**). We will include results for additional LLMs and configurations in the final version (as suggested by `Reviewer T7ko`) to highlight clearer gaps and more concrete conclusions. Many studies have shown that LLM-as-a-Judge can be a reliable alternative to human evaluation.
> >
> > 5. The **case study** was designed  to demonstrate the `challenging examples` or *"error cases"* identified through manual inspection (`Lines 961-963`); it does not reflect all samples.
> >
> > 6. Regarding verifiers: for Opinion Faithfulness (OF), we explain in `Lines 237–240` that we extract a set of opinions from retrieved evidence that share the same aspect and sentiment. Since the comparisons occur within sentiment-aligned sets, contrasting terms like “large” vs. “small” do not co-occur in opinion set for verification.
> >
> > Lastly, we respectfully note that this submission already introduces **(i)** a new benchmark dataset (OpinioBank), **(ii)** a modular, training-free RAG framework (OpinioRAG), and **(iii)** a verification module based on AOS triplets. Gaps are inherent to research and invite further exploration by the community.
> >
> > **Your review has helped us refine the paper, and we are fully committed to incorporating your suggestions in the revised version. Thank you again for your engagement and for giving us the opportunity to clarify.**

---

> > > ### Comment · Reviewer_Pz2B · 2025-06-10
> > >
> > > Thank you for your response. Our discussions have addressed several of my questions and I've updated the score.

---

> > > > ### Author Response · Authors · 2025-06-10
> > > > **Thank you!**
> > > >
> > > > Thank you very much for **recommending our paper for acceptance**. We sincerely appreciate your engagement throughout the review process and are grateful for the opportunity to improve our work through your feedback.

---

### Official Review · Reviewer_T7ko · 2025-05-12

**Rating:** 6
**Confidence:** 4
**Ethics Flag:** 2

**Summary:**

The paper introduces a new large-scale dataset for RAG-based opinion summarization.  The dataset includes 500 hotels, with an average of 1500 reviews from TripAdvisor per hotel, which are paired with an expert review from Oyster, serving as a reference summary. In addition, query terms are manually extracted from the expert summary. The proposed framework retrieves the top k sentences matching the given query and applies an LLM to generate query-specific “highlights” in key-point style. An automatic verification method for the highlight given the retrieved evidence is also proposed, based on aspect-opinion-sentiment decomposition. Experimental results demonstrate that the RAG-based approach outperforms feeding all the reviews to long-context models.

**Ethics Concerns Details:**

My concern is about the data licensing - waiting for clarification from the authors (see Questions for Authors).

**Questions To Authors:**

Can you please clarify the data licensing issue mentioned above, regarding the benchmark data scraping from TripAdvisor and Oyster and how it adheres to their license and terms of use? --> answered in the rebuttal.

**Reasons To Accept:**

* The paper introduces a benchmark for opinion summarization comprising long-form reviews, which is significantly larger and more diverse than previous benchmarks, and may be useful for future research in this area.

* Matching the reviews from TripAdvisor with expert summaries from Oyster is a nice idea, that allows automatic generation of a large-scale dataset.

**Reasons To Reject:**

Updated following the rebuttal.

* The RAG based framework and the evaluation, which get the query terms as input, greatly simplifies the task, but the more realistic and challenging task, which is not addressed by this work, is end-to-end summary generation. --> while I still think that generating summaries without the extra input of query terms is the more realistic and challenging task, I accept the authors' point that user-directed summarization is a valid and interesting task by itself.

* The paper should be re-organized to be more self-contained. Important parts such as experimental setup and related work only appear in the appendix, making the paper difficult to follow. It would also help to move Figure 1 to the introduction, to have an overview of the framework early on. --> the authors promised to address this in the camera-ready version

* While the authors claim in the ethics statement that all the collected data complies with the CC-BY-NC-SA license, the benchmark was obtained by scraping commercial websites. The authors should explain the licensing issue and how their use of the data complies with the websites’ term of use. --> the authors clarified this point.

* Using the expert review as a summary of user reviews may be a reasonable approximation, but it should be noted and discussed in the paper that the expert review and the user reviews may not necessarily agree.

* Given the known limitations of lexical and lexical-embedding  methods such as ROUGE and BERTScore, it would be much more convincing to see evaluation results based on a reference-based LLM judge. --> the authors partially addressed this point by providing LLM-judge based results for some of the configurations. I think they should be extended to all the tested configurations.

---

> ### Author Response · Authors · 2025-06-02
> **Authors' Response (Part 1/2)**
>
> Thank you for raising this important point—it allows us to clarify our adherence to data licensing norms. Website Terms of Use policies are often broad and general, and typically do not explicitly address non-commercial academic research usage. Therefore, we followed several precautionary steps to ensure compliance and alignment with community standards.
>
> ### **Legal and Community Context**
>
> All data collected from TripAdvisor and Oyster is **publicly available**. Scraping publicly accessible content from the **web is legal** [[Link](https://techcrunch.com/2022/04/18/web-scraping-legal-court/)]. For example, in the US, case [hiQ Labs, Inc. v. LinkedIn Corp.](https://en.wikipedia.org/wiki/HiQ_Labs_v._LinkedIn), courts—including the Supreme Court—ruled that scraping **public data does not violate** the Computer Fraud and Abuse Act (CFAA). Similar [legal interpretations](https://digital-strategy.ec.europa.eu/en/policies/digital-services-act-package) exist in the EU, UK, and China. This legal precedent has enabled large-scale corpus like **Common Crawl**, which underpin the development of modern LLMs.
>
>
> Given that Terms of Use do not provide explicit restrictions for non-commercial academic use, our data collection approach aligns with [established practices](https://aclrollingreview.org/responsibleNLPresearch/) in NLP research that involve publicly available web content. Prior *ACL/NeurIPS publications, such as **SPACE** (Angelidis et al., 2021) and **OpoSum+** (Amplayo et al., 2021), similarly use data from TripAdvisor and Amazon for opinion summarization.
>
> ### **Steps We Took**
>
> 1. We checked the `robots.txt` files (https://moz.com/learn/seo/robotstxt) of each website to verify crawl permissions (allow or ‘no restrictions’) according to the Robots Exclusion Protocol (REP), which is widely treated as the technical standard for crawl permissions. For example, TripAdvisor’s **robots.txt** (https://www.tripadvisor.ca/robots.txt) explicitly includes `‘Allow: /UserReview$’`, indicating that user review pages are crawlable.
> 2. We conducted scraping at a reasonable rate to avoid triggering server-side protections or causing harm (`Lines 350–352`), and we included HTTP User-Agent headers in all requests.
> 3. We did not collect any **Personally Identifiable Information (PII)**, consistent with GDPR and CCPA. This is elaborated in `Lines 353–358`.
> 4. We clearly specify that the dataset will be released under a **CC BY-NC-SA 4.0** license (`Lines 346–350`), ensuring it is strictly for non-commercial research purposes.
> 5. Our OpinioBank is a lightweight dataset with 500 expert-paired samples, intended solely for evaluation and benchmarking—thus significantly limiting both the volume of collected content and any potential commercial relevance (`Table 1`).
>
> In the revised version, we will add further clarification to the Ethics Statement. We believe this response comprehensively addresses your concern and reaffirms that our data usage is both legal and aligned with accepted academic practices. Please let us know if you have any additional questions or feedback.
>
> -----------------------
>
> ### **Improving Readability**
>
> We completely agree with your suggestion. In the camera-ready version (*if accepted*) given an extra page, we will reorganize the paper to improve clarity and self-containment by moving key elements from the Appendix into the main paper. We will also relocate Figure 1 to the Introduction to provide an early and intuitive overview of the framework.
>
> -----------------------
>
> ### **End-to-end Summary Generation**
>
> Our approach, OpinioRAG is indeed designed to simplify the task through a modular RAG-based setup. However, we also address the more challenging end-to-end summary generation setting using long-context LLMs with OpinioBank as baselines.
>
> As shown in **Table 2** and described in `Lines 838–844` (with prompt in **Figure 4**), we evaluate models where all user reviews are concatenated and passed as input—without any intermediate retrieval. To further respond to **Reviewer Pz2B's** question, we also evaluated *two additional* open-source long-context models in this setting and will include these results in the revised Table 2. This may have been unclear because some of these details currently appear in the Appendix. We will move the relevant discussion into the main body.

---

> > ### Author Response · Authors · 2025-06-02
> > **Authors' Response (Part 2/2)**
> >
> > ### **Empirical Validation of Pairing Quality**
> >
> > While we acknowledge that expert reviews and user reviews may not always perfectly align, we explicitly designed a **verification mechanism** to ensure reliable pairing (`Lines 130–135`). We also compare OpinioBank against **fully human-annotated datasets** (Appendix A2, `Lines 784–790`) and show in **Table 6** that our dataset achieves superior alignment with the underlying user reviews. This suggests that our query-based filtering and verification pipeline is both principled and empirically effective.
> >
> > -----------------------
> >
> > ### **Evaluation via LLM-as-a-Judge**
> >
> > Motivated by your suggestion, we conducted a comprehensive *LLM-as-a-Judge evaluation* using GPT-4o across all samples in our OpinioBank dataset. We assessed the best-performing open-source and closed-source models under the OpinioRAG framework (**Table 2**), spanning multiple retrieval strategies and Top-k configurations. Each summary was evaluated on a **5-point Likert scale** across the following quality dimensions.
> >
> > This evaluation directly addresses the motivation in your review, providing a structured and fine-grained assessment of generation quality. As shown in the table below, the Claude-3.5-haiku + Dense + Top-10 configuration consistently outperforms others, particularly in factual grounding and pragmatic coherence—highlighting the advantage of closed-source models. We hope this evaluation **reinforces our contributions**, and we sincerely appreciate your valuable suggestion.
> >
> > **Evaluation Criteria:**
> >
> > 1. **Aspect Relevance (AR):** Does the system summary cover the same topics or facets as the expert summary (even if worded differently)?
> > 2. **Non-Redundancy (NR):** Are aspects mentioned only once? Are key points repeated or paraphrased redundantly?
> > 3. **Sentiment Agreement (SA):** Is the tone (positive or negative) about aspects consistent between the summaries?
> > 4. **Opinion Faithfulness (OF):** Are the factual or evaluative claims in the system summary grounded in the expert summary?
> > 5. **Overall Usefulness (OU):** Would the system summary help a potential customer make a reasonable decision?
> >
> > | Model Name       | Retrieval | TopK |  AR   |  NR   |  SA   |  OF   |  OU   |
> > |------------------|-----------|------|-------|-------|-------|-------|-------|
> > | Llama-3.1-8B      | BM25      | 5    | 3.17  | 3.83  | 2.92  | 2.82  | 3.12  |
> > |                  |           | 10   | 3.26  | 3.81  | 2.99  | 2.87  | 3.20  |
> > |                  | Dense     | 5    | 3.23  | **3.94** | 3.04  | 2.89  | 3.19  |
> > |                  |           | 10   | 3.31  | 3.84  | 3.10  | 2.93  | 3.26  |
> > | Claude-3.5-haiku | BM25      | 5    | 3.20  | 3.82  | 2.94  | 2.90  | 3.17  |
> > |                  |           | 10   | 3.25  | 3.73  | 3.02  | 2.95  | 3.22  |
> > |                  | Dense     | 5    | 3.28  | 3.85  | 3.08  | 2.95  | 3.23  |
> > |                  |           | 10   | **3.38** | 3.73  | **3.13** | **2.98** | **3.33** |
> >
> > -----------------------
> >
> > We hope this comprehensive response addresses **all of your thoughtful concerns**. We truly appreciate your critical engagement, which has helped us further refine our work. If you have any remaining questions, we would be glad to address them.

---

> > > ### Comment · Reviewer_T7ko · 2025-06-05
> > >
> > > Thank you for your response! I appreciate your detailed explanation regarding data licensing, which addressed my concern.
> > >
> > > I was also pleased to see the new LLMaJ results. I think they should be expanded to include also the non-retrieval configurations, to support the claim that the retrieval-based approach is better.
> > >
> > > Regarding the paper's re-organization - I'm glad you plan to address this. However, I think it's a major revision that should be reviewed before acceptance.
> > >
> > > Regarding the end-to-end setting: your non-retrieval configurations still get the user queries as an input, which guides the summarizer, as opposed to a standard summarization or key point analysis setting where user queries are not provided. It is actually not difficult to test this in your setting, by removing user queries from the LLM's input.
> > >
> > > Given the above, I'm leaning toward "revise and resubmit", and will maintain my current score.

---

> > ### Author Response · Authors · 2025-06-05
> >
> > Thank you for your thoughtful response and for **acknowledging** that our clarifications addressed your **major concerns**, including data licensing. We're also glad you **appreciated the inclusion** of the new LLM-as-a-Judge results.
> >
> > -------------------
> >
> > ### **On Paper Reorganization**
> >
> > **`As you noted in your original review`**, the reorganization mainly involves moving the experimental setup, related work into the main paper from the Appendix and Figure 1 to the Introduction. We agree this is a **straightforward revision** and plan to incorporate it fully into the camera-ready version. We see this as a **minor structural change** without affecting core content (given an **extra page** in the final version).
> >
> > -------------------
> >
> > ### **On End-to-End Setting**
> >
> > Our choice to include user queries in the end-to-end setting was **intentional**, to enable a controlled, **apple-to-apple comparison** with the RAG-based setup. Since our focus is on `user-centric opinion highlight generation`, incorporating user intent via queries like *“yoga classes”* or *“room bathroom”* is essential. Additionally, we aim to analyze how long-context models cope with the `“lost-in-the-middle”` issue in this realistic setting.
> >
> > While we acknowledge that query-free summarization is an interesting extension, we believe this is a broader design question rather than a missing component. Our current design and experiments are **well-justified** for the research scope we intended to pursue in this paper.
> >
> > -------------------
> >
> > ### **Overall Contributions**
> >
> > We respectfully note that this submission (**`already 35 pages`**) presents a new  benchmark dataset (**OpinioBank**), a modular, training-free RAG framework (**OpinioRAG**), and a novel structured verification module based on AOS triplets. We believe these contributions are **substantial and impactful** for the summarization and retrieval-augmented generation communities.
> >
> > Given the clarifications above, we hope you consider **revisiting your score**. We genuinely appreciate your engagement and welcome any further discussions.

---

> > > ### Author Response · Authors · 2025-06-06
> > > **Request for Score Reconsideration**
> > >
> > > We respectfully urge you to **reconsider** your score. We have addressed **all major concerns raised in your review**. Moving the experimental setup, related work to the main body and Figure 1 to the Introduction—as you suggested—is a **`straightforward revision`**, and we are **fully committed** to do this in the camera-ready version using the extra page allowance.
> > >
> > > The **`“revise and resubmit”`** option is only supported under rolling review platforms like ACL ARR. Given that our work introduces a new dataset (**OpinioBank**), a training-free **OpinioRAG** framework, a novel AOS-based verification module, and comprehensive evaluations—we believe the current version already offers **substantial and timely contributions** deserving of acceptance.

---

> > > > ### Comment · Reviewer_T7ko · 2025-06-08
> > > >
> > > > Dear authors, I updated my review based on our discussion, and raised my rating from 4 to 6.

---

> > ### Author Response · Authors · 2025-06-09
> > **Thank you!**
> >
> > Thank you very much for **recommending our paper for acceptance**. We acknowledge your revised review and will incorporate the remaining points in the final version. These are straightforward revisions, and we are fully committed to addressing them.

---

### Official Review · Reviewer_3mxc · 2025-05-14

**Rating:** 7
**Confidence:** 4
**Ethics Flag:** 1

**Summary:**

The paper presents a RAG-based framework for opinion summarisation called OpinioRAG. The authors create datasets of hotel reviews from two sources, and label sentences with 'queries' that can aid retrieval. They then compare OpinioRAG (with multiple LLMs and two retriever methods) alongside multiple prompting baselines on long context LLMs.

**Questions To Authors:**

- Aspect, Opinion, Sentiment triplets  (Line 198) have been described way earlier than 2023 by Bing Liu et al. The authors are requested to please correct the citation.

**Reasons To Accept:**

- The paper is written well and easy to follow.

- The idea of opinion highlights from a large number of reviews influences several aspects of the paper: the additional labeling for queries, exploration of structured reviews and different types of retrievers.

- The experimentation is great - lots of baselines,  retriever methods and sentiment highlights.

In that sense, the paper takes upon a challenging problem (opinion highlights), clearly presents its contributions (Table 1), and compares the proposed approach against multiple models (Table 2).  The discussion section identifies potential future research directions which look feasible on the top of this work.

**Reasons To Reject:**

The authority of opinion highlights themselves is questionable. However, this is a minor concern, in my opinion.

---

> ### Author Response · Authors · 2025-06-02
> **Authors' Response**
>
> We sincerely thank the reviewer for the thoughtful and encouraging feedback. We are glad that you appreciated the writing quality, experimental rigor, and structured framework design of our paper.
>
> -----------------------
>
> ### **Regarding the authority of opinion highlights**
>
> The authority of opinion highlights is grounded in **reviewer-side metadata**, such as the *total number of reviews posted*, *the number of distinct cities covered*, and the *cumulative number of helpful votes received*. As discussed in **Section 6** (`Lines 330-337`), these  signals reflect key aspects of  reviewers’ experience, trustworthiness, and credibility. These attributes are available in the dataset and can be leveraged to inform generation or alignment, for example through automatic filtering or weighting of input reviews. Although underexplored in prior work, we believe this incorporation of opinion holder information is an important contribution of our dataset and opens promising directions for future research.
>
> -----------------------
>
> ### **Regarding missing citations**
>
> You're absolutely right. Aspect–Opinion–Sentiment triplets were introduced well before 2023, notably in  the seminal work by Bing Liu (Hu & Liu, 2004),  which we cite in the introduction (`Lines 29–31`).  We will also add a citation to Sentiment Analysis and Opinion Mining (Liu, 2012) at Line 198 along with (Hu & Liu, 2004) to properly acknowledge his broader contribution. If there are additional specific references you recommend, we would be happy to include them as well.
>
> -----------------------
>
> We greatly appreciate your positive assessment and constructive suggestions. Your support has already been very encouraging, and any additional support you might consider would go a long way in enhancing the visibility and recognition of this work within the community.

---

> > ### Comment · Reviewer_3mxc · 2025-06-03
> > **thnx**
> >
> > Thanks for the response, and best wishes for the paper

---

### Decision · Program_Chairs · 2025-07-08

**Decision:**

Accept

**Comment:**

The paper introduces a new large-scale dataset for RAG-based opinion summarization. The dataset includes 500 hotels, with an average of 1500 reviews from TripAdvisor per hotel, which are paired with an expert review from Oyster, serving as a reference summary. In addition, query terms are manually extracted from the expert summary. The proposed framework retrieves the top k sentences matching the given query and applies an LLM to generate query-specific “highlights” as  pros and cons. An automatic verification method for the output summary given the retrieved evidence is also proposed, based on aspect-opinion-sentiment decomposition. Experimental results demonstrate that the RAG-based approach outperforms feeding all the reviews to long-context models (and well-known baselines in the opinion summarization literature).

The paper could benefit from reorganization of its content as suggested by reviewer T7ko. The authors mostly addressed the reviewers' questions and added LLM-as-a-judge evaluation in the rebuttal stage. It would be good to see human evaluation in the camera-ready, I know the long context is prohibitive; however, evaluation could be conducted by giving participants the sentences retrieved using the query term or a sample thereof if they are too many. Most of the papers you reference in Table 1 conduct human evaluation.

**This paper went through ethics reviewing. Please review the ethics decision and details below.**
Decision: All good, nothing to do  or only minor recommendations